# IL-2/IL-7-inducible factors pioneer the path to T cell differentiation in advance of lineage-defining factors

Sarah L Bevington[1,*] [iD], Peter Keane[1] [iD], Jake K Soley[1], Saskia Tauch[1], Dominika W Gajdasik[2], Remi Fiancette[2], Veronika Matei-Rascu[2], Claire M Willis[2], David R Withers[2,**] [iD] & Peter N Cockerill[1,***] [iD]

## Abstract

When dormant naïve T cells first become activated by antigen-presenting cells, they express the autocrine growth factor IL-2 which transforms them into rapidly dividing effector T cells. During this process, hundreds of genes undergo epigenetic reprogramming for efficient activation, and also for potential reactivation after they return to quiescence as memory T cells. However, the relative contributions of IL-2 and T cell receptor signaling to this process are unknown. Here, we show that IL-2 signaling is required to maintain open chromatin at hundreds of gene regulatory elements, many of which control subsequent stimulus-dependent alternative pathways of T cell differentiation. We demonstrate that IL-2 activates binding of AP-1 and STAT5 at sites that can subsequently bind lineage-determining transcription factors, depending upon what other external factors exist in the local T cell environment. Once established, priming can also be maintained by the stroma-derived homeostatic cytokine IL-7, and priming diminishes if *Il7r* is subsequently deleted *in vivo*. Hence, IL-2 is not just a growth factor; it lays the foundation for T cell differentiation and immunological memory.

**Keywords** differentiation; IL-2; memory T cell; priming; transcription factor
**Subject Categories** Immunology; Signal Transduction
**The EMBO Journal (2020) 39: e105220**

## Introduction

The adaptive immune system in mammals relies on T lymphocytes which express a wide range of immune modulators and defense molecules when they respond to a specific foreign antigen (Ag). Although T cells exit the thymus as mature T cells, they remain fixed in a relatively unresponsive state as naïve T cells until they undergo further differentiation triggered by encounters with antigen (Ag). CD4 helper T cells only acquire their true potential within the immune system after their commitment to differentiate to different types of effector T cells equipped to respond to different classes of pathogens presenting in different environments.

When naïve T cells are activated by Ag-presenting cells for the first time via T cell receptor (TCR) and CD28 co-receptor signaling, they rapidly turn on expression of the autocrine growth factor interleukin-2 (IL-2) and its receptor IL-2Rα, and undergo extensive genome-wide chromatin reorganization during their transformation into rapidly dividing T-blast cells. This process is driven by the TCR/CD28-inducible transcription factors (TFs) NFAT, AP-1, NF-κB, and EGR1 (Yukawa *et al*, 2019), and supported by the IL-2-inducible TFs AP-1 and STAT5 (Liao *et al*, 2011a; Ross & Cantrell, 2018). Activation of STAT5 is mediated by phosphorylation via JAK1/3 which gets recruited to the IL-2 receptor in response to binding of IL-2 (Miyazaki *et al*, 1994), whereas AP-1 induction involves transcriptional activation via RAS and kinase signaling, which is also activated by TCR signaling. These TFs, in turn, activate hundreds of inducible immune response genes and create thousands of open chromatin regions identified as DNase I hypersensitive sites (DHSs) which represent enhancer and promoter elements that activate the expression of inducible immune response genes. Once TCR signaling ceases and inducible TFs are down-regulated, the majority of these sites disappear. However, ~ 3,000 regions are stably maintained as epigenetically primed DHSs (pDHSs) which function to preserve transcriptional and immunological memory once a T cell loses contact with an Ag-presenting cell (Bevington *et al*, 2016). These pDHSs establish islands of active chromatin encompassing the inducible promoters and enhancers, thereby enabling the TCR-inducible genes to be rapidly reactivated more efficiently when effector T cells and memory T cells re-encounter Ag (Fig 1A; Bevington *et al*, 2016, 2017a,b). Furthermore, the majority of these pDHSs do not function as classical enhancers because their presence

1 Institute of Cancer and Genomic Sciences, College of Medical and Dental Sciences, University of Birmingham, Birmingham, UK
2 Institute of Immunology and Immunotherapy, College of Medical and Dental Sciences, University of Birmingham, Birmingham, UK
 *Corresponding author. Tel: +44 1214144065; E-mail: s.l.bevington@bham.ac.uk
 **Corresponding author. Tel: +44 1214148516; E-mail: d.withers@bham.ac.uk
 ***Corresponding author. Tel: +44 1214146841; E-mail: p.n.cockerill@bham.ac.uk

has no impact on steady-state levels of transcription, or on the activity of reporter genes in transfection assays (Bevington *et al*, 2016, 2017a). Hence, pDHSs make it possible to maintain inducible genes in a poised non-transcribed state while also enabling rapid reactivation. pDHSs are stably maintained in both rapidly dividing T-blast cells *in vitro* and in quiescent memory T cells *in vivo* by binding constitutively expressed TFs such as RUNX1 and ETS1. However, many pDHSs also have binding sites for IL-2-inducible STAT5 and AP-1 proteins, suggesting a potential additional role for IL-2 in the formation and/or maintenance of these sites. Importantly, while the initial activation and any subsequent reactivation of immune response genes in T cells is highly dependent on TCR signaling, the maintenance of the proliferative response and transcriptional memory can be supported by just IL-2 in the absence of Ag (Bevington *et al*, 2016).

IL-2 binds to its receptor via the IL-2-specific chain IL-2Rα but signals via the common gamma chain (γc) which is shared by the receptors for five other cytokines (IL-4, IL-7, IL-9, IL-15, and IL-21) (Liao *et al*, 2011a; Ross & Cantrell, 2018; Leonard *et al*, 2019) which are important for the action and response of many cells in the immune system (Read *et al*, 2016; Ross & Cantrell, 2018). IL-2, IL-7, and IL-15 are collectively integral to the establishment, homeostasis, and maintenance of memory T cells (Raeber *et al*, 2018). Although the role of IL-2 as a T cell growth factor for recently activated T cells has been known for many decades (Morgan *et al*, 1976), it also plays an important role during memory T cell formation since loss of IL-2 signaling early during T cell activation impairs subsequent memory T cell recall (Williams *et al*, 2006; Pepper *et al*, 2011). Furthermore, IL-2 is also needed early during T cell activation for the later expression of IL-7Rα, which is required for the formation of the memory T cell pool (Dooms *et al*, 2007).

Following their initial activation, CD4 T cells are normally destined to differentiate to become specialized effector helper T (Th) cells, such as Th1, Th2, or Th17, which each express a different specific subset of immune response genes (Zhu *et al*, 2010). This process is driven by different cytokines which are expressed by various cell types in response to different pathogens (Fig 1B). Cytokine activity triggers signaling via different classes of STAT proteins and activates expression of alternative lineage-determining TFs, depending upon the circumstances: Viral and bacterial infections typically result in local expression of IL-12, which triggers signaling via STAT4 and expression of the TF T-BET (*Tbx21*) which drives Th1 differentiation and expression of IFNγ (Szabo *et al*, 2003); parasitic infections typically result in local expression of IL-4, which triggers signaling via STAT6 and expression of the TF GATA3 which drives Th2 differentiation and expression of IL-4 and IL-5 (Ansel *et al*, 2006); mucosal membrane infections typically result in local expression of IL-6, which triggers signaling via STAT3 and expression of the TF RORγt (*Rorc*) which drives Th17 differentiation and expression of IL-17 (Littman & Rudensky, 2010). It has previously been widely presumed that factors such as T-BET, GATA3, and RORγt function as true pioneer factors, much like the archetypal pioneer factor FOXA1 (Cirillo *et al*, 2002), and are able to initiate the process of opening up previously inaccessible chromatin domains, thereby triggering lineage-specific gene expression.

While it is accepted that epigenetic modifications and TF action play crucial roles in T cell activation, differentiation, and memory (Abdelsamed *et al*, 2018), little is known regarding the extent to which γc cytokines contribute to the establishment or maintenance of epigenetic modifications acquired by previously activated T cells, and for the priming the cytokine-dependent induction of terminal T cell differentiation. STAT5, AP-1, and IL-2 have been implicated in the regulation of T cell activation and polarization in Th1 (Shi *et al*, 2008; Liao *et al*, 2011b; Zeng *et al*, 2016), and Th2 cells (Ben-Sasson *et al*, 1990; Le Gros *et al*, 1990; Rincon & Flavell, 1997; Hartenstein *et al*, 2002; Zhu *et al*, 2003; Cote-Sierra *et al*, 2004; Yamane *et al*, 2005; Betz *et al*, 2010; Ross & Cantrell, 2018) with enrichment of AP-1 and STAT motifs at distal elements that were also enriched with motifs for lineage-specific transcription factors (Zeng *et al*, 2016). Finally, IL-2 contributes directly to Th1 differentiation by promoting expression of IL-12Rβ1 and IL-12Rβ2, and to Th2 differentiation by promoting expression of IL4-Rα and by priming the locus control region (LCR) of the locus encompassing genes encoding the Th2 cytokines IL-4, IL-13, and IL-5 (Liao *et al*, 2008, 2011b).

To determine the role of IL-2 in this process, we examined the *in vitro* responses of recently activated T cells to short-term withdrawal of IL-2. We show that IL-2 is required to maintain 1,000 pDHSs bound by STAT5 and AP-1 in recently activated and rapidly proliferating CD4 T-blast cells and that most of these sites also represent DHSs that respond subsequently to lineage-defining cytokines and TFs. Thus, IL-2 signaling does not just act early to support transformation of quiescent naïve T cells but plays a genome-wide role in establishing lineage-specific patterns of gene regulation long before the first appearance of lineage-specific factors.

# Results

## IL-2 is required to maintain a subset of DHSs in recently activated T-blast cells

To identify specific roles for IL-2 signaling in T cell development, we investigated its contribution to maintaining pDHSs in recently activated proliferating CD4 T-blast cells ($T_B$). We performed multiple genome-wide analyses of actively dividing T-blast cells immediately after a cycle of transient TCR activation, followed by culture with IL-2, but before the cells are polarized toward different T helper cell subsets in the presence of lineage-defining cytokines and TFs. Hence, we are studying very early reprogramming events that take place in Th0 cells prior to terminal differentiation which can take several additional days or weeks (Agarwal & Rao, 1998).

To investigate the role that this pan T cell-specific cytokine plays in maintaining priming in Th0 cells, we employed an *ex vivo* T-blast transformation model system (Fig 1C). In this model, TCR signaling is activated in CD4 naïve T cells ($T_N$) using concanavalin A (ConA) for 40 h, and following transformation, rapidly proliferating T-blast cells are maintained in culture and undergo several cell divisions in the presence of IL-2 in place of ConA. In the presence of just IL-2, the transcriptional profile of TCR-inducible genes in $T_B$ cells returns to steady-state levels more similar to that seen in $T_N$ cells prior to transformation. However, many of the recently activated immune response genes still maintain epigenetic priming in the absence of TCR/CD28 signaling, thereby enabling much faster reactivation by TCR signaling than occurs in $T_N$ cells (Bevington *et al*, 2016). To assess the role of IL-2 in the maintenance of this epigenetic priming,

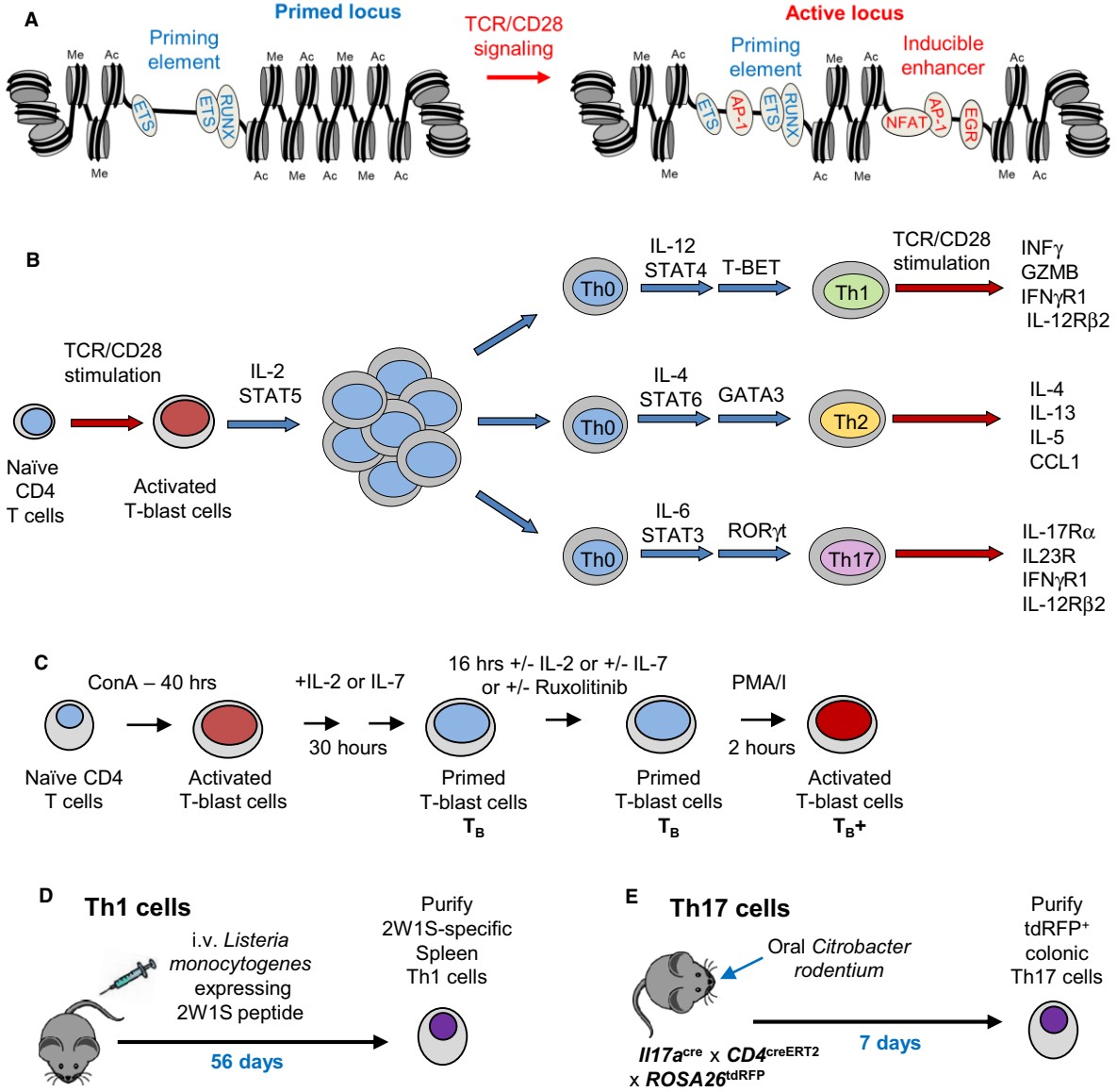

**Figure 1. CD4 T cell activation and differentiation.**

A   Model of an epigenetically primed inducible locus before and after reactivation of TCR/CD28 signaling.

B   Schematic of alternative pathways of cytokine-induced helper T cell differentiation in giving rise to Th1, Th2, or Th17 subtypes of helper T cells.

C   Schematic of T-blast cell transformation. Purified CD4 T cells were activated with ConA for 40 h and maintained in IL-2 or IL-7 for 30 h as $T_B$. Cells were cultured for a further 16 h ± IL-2, IL-7, or Ruxolitinib. Cells were re-stimulated ($T_B$+) with 20 ng/ml PMA and 2 mM calcium Ionophore (PMA/I) for 2 h.

D, E   Schematics of the procedures for generating Th1 cells in response to intravenous infection of bacteria (D) and Th17 cells by oral infection with bacteria (E).

recently transformed $T_B$ cells were first cultured with IL-2 for 30 h and then for 16 h in the absence of IL-2. Removal of lL-2 from the cultures led to a significant reduction in *Il2ra* mRNA expression, suggesting that the contribution from any autologous IL-2 will be limited (Fig EV1A).

To measure levels of epigenetic priming, we performed DNase-Seq to identify all open chromatin regions containing active regulatory elements. CD4 $T_B$ cells were cultured in the presence or absence of IL-2 ($T_B$ IL-2 and $T_B$ IL-2nil), non-apoptotic live cells were purified (Fig EV1B) before DNase I treatment, and the DHSs

were ranked according to the fold change in the DNA sequence tag count of peaks in $T_B$ IL-2 compared to $T_B$ IL-2nil (Figs 2A and EV1C). In the absence of IL-2, 1,000 IL-2-dependent DHSs (IL-2 pDHSs) were reduced in size by at least twofold in each of 2 replicates (Figs 2B and EV1C, Table EV1). Many of these IL-2-dependent sites (286) were included within the group of 2,882 primed DHSs that we previously showed were maintained *in vivo* in memory phenotype T cells (Bevington *et al*, 2016; Fig EV1D). Indeed, in the absence of IL-2 there was a substantial reduction in average DNase I signal for the whole population of 2,882 *in vivo*-defined pDHSs

compared to a control group of 2,882 invariant DHSs shared by $T_B$ and $T_N$ (Fig EV1E).

To investigate whether the IL-2 pDHSs defined *in vitro* were also stably propagated in previously activated T cells *in vivo*, we looked to see whether they were present in purified *in vivo*-differentiated T cells. For this purpose, we generated (i) Ag-specific Th1 memory T cells by immunizing mice with attenuated *Listeria monocytogenes* expressing the 2W1S peptide (*Lm*-2W1S), and purifying the 2W1S-specific T cells 56 days later (Fig 1D; Pepper *et al*, 2010), and (ii) bonafide Th17 cells by orally infecting *Il17a*^cre × ROSA26^tdRFP+ mice with *Citrobacter rodentium* and sort purifying colonic tdRFP+ fate-mapped CD4 T cells 7 days later (Fig 1E; Hirota *et al*, 2011). We also made use of published data sets from Th2 cells (Shih *et al*, 2016), regulatory T cells (Treg) (Garg *et al*, 2019), and T follicular helper cells (Tfh) (Chen *et al*, 2020). Significantly, we were able to detect open chromatin regions corresponding to 26–52% of the 1,000 IL-2 pDHSs in each of these five subsets of differentiated CD4 T cells (Fig 2A and C). Furthermore, 65% of the 1,000 IL-2 pDHSs were present in at least one of the Th1, Th2, or Th17 subsets (Fig 2D), indicating that IL-2-dependent pDHSs play additional roles in supporting Th differentiation, and are potentially also involved in the maintenance of priming by homeostatic cytokines such as IL-7 and IL-15 *in vivo*.

To investigate the role of JAK/STAT signaling from IL-2 to the primed DHSs, we treated cells with the JAK1/2 inhibitor Ruxolitinib (INCB018424). $T_B$ cells were cultured for 30 h in IL-2 before addition of 1 μM Ruxolitinib for a further 16 h (Fig 1C). Ruxolitinib treatment was sufficient to suppress 36% of the IL-2 pDHSs compared to the DMSO control, thereby identifying a subset of IL-2-dependent pDHSs that rely on STAT5 activity and not just AP-1 (Figs 2A, and EV1F and G).

The IL-2 dependence at pDHSs was exemplified by multiple regions that we previously identified as primed in recently activated T-blast cells (Bevington *et al*, 2016), including the Th2-specific LCR (Fields *et al*, 2004) within the *Rad50* gene (Fig 2E), which regulates an archetypal Th2-specific cytokine gene cluster that includes *Il4, Il13,* and *Il5* (Bevington *et al*, 2017b), the *Tnfsr9* locus, and the IL-2-regulated *Cish* locus region (Li *et al*, 2017). For these loci, IL-2 pDHSs were present in CD4 $T_B$ IL-2 and CD4 $T_B$ DMSO cells, and they were reduced upon IL-2 removal and by Ruxolitinib treatment. Furthermore, an IL-2 and JAK1-dependent DHS was detected in the *Il17a* locus, an archetypal Th17 gene (Fig EV1H). Although this DHS did not qualify as one of the 1,000 high confidence IL-2 pDHSs, due to the peak being below the threshold for detection in the second $T_B$ IL-2 DNase-Seq replicate, it was clearly present in the DMSO control used to define the Ruxolitinib sensitivity of this site, and in Th17 cells (Fig EV1H). Notably, as exemplified by the CD3 gene cluster and *Tbp* loci, removal of IL-2 or treatment with Ruxolitinib did not result in a global reduction in DNase I hypersensitivity (Fig EV1I). In summary, our data show that IL-2 signaling is needed to maintain epigenetic priming at recently reprogrammed committed gene loci prior to terminal T cell differentiation.

### IL-2-dependent DHSs bind IL-2-inducible factors

To define potential molecular mechanisms responsible for maintaining epigenetic and transcriptional priming at pDHSs, we sought evidence for specific priming factors binding to pDHSs so as to keep them in an open conformation poised for rapid reactivation. Using HOMER, we performed a *de novo* DNA motif search for enriched TF consensus DNA-binding sequences within the IL-2 pDHSs, which revealed that they were highly enriched in binding sites for the JAK1/2-inducible factor STAT5 and the MAPK-inducible factor AP-1 (34 and 37% respectively) (Fig 3A).

We previously established that although the expression of most AP-1 genes, including *Fos, Fosb, Jun, and Junb,* is highly TCR-dependent in T cells, *Jund* mRNA is expressed at substantial levels in the presence of just IL-2 (Bevington *et al*, 2016). In the present study, we confirmed that the expression of *Jund* mRNA and JUND protein is IL-2-dependent, as is the phosphorylation of STAT5 (Fig 3B and C). Furthermore, the locations of the STAT and AP-1 consensus motifs were highly concentrated within the most strictly IL-2-dependent fraction of DHSs (Fig 3D). DNA motifs for the constitutively expressed RUNX and ETS family TFs were also highly enriched within the IL-2 pDHSs (Fig 3A), but these motifs were not restricted to the IL-2-dependent DHSs (Fig 3D).

To confirm the identities of specific members of TF families bound at consensus motifs in pDHSs, we performed chromatin immunoprecipitation assays (ChIP-Seq) for the IL-2-inducible TFs JUND and STAT5 in $T_B$ IL-2. Two replicate samples were used to confirm that both transcription factors were enriched at the IL-2 pDHSs (Figs 3E and EV2A). We also performed a single ChIP-Seq assay for JUND and STAT5 in $T_B$ IL-2nil just to confirm that their binding was indeed lost in parallel with the loss of pDHSs in the absence of IL-2 (Figs 3E and EV2B). Curiously, strong STAT5 binding was also seen at the shared IL-2-independent DHSs in the absence of IL-2 (Fig 3E). Hence, it is likely that significant levels of binding of presumably un-phosphorylated STAT5 still occur throughout the genome in the absence of IL-2 (Fig 3E). Others have also previously observed that STAT5 phosphorylation is not essential for it to associate with DNA (Park *et al*, 2016). Examples of reduced STAT5 and JUND binding when IL-2 is absent are observed at primed regions within TCR-inducible immune response genes such as *Il13, Gzmb,* and *Il24* (Fig 3F). As an additional control, we also demonstrated that overall levels of STAT5, ETS1, and RUNX1 did not change in response to IL-2 (Fig EV2C). Taken together, these data suggest that IL-2-regulated proteins are required to maintain a subset of pDHSs in recently activated T cells.

### Loss of IL-2 signaling correlates with suppression of inducible DHSs at previously activated genes

To investigate whether IL-2 not only maintains primed DHSs but also regulates inducible immune response genes, we activated TCR signaling in $T_B$ IL-2 and $T_B$ IL-2nil with PMA and calcium ionophore (PMA/I) for 2 h ($T_B$+) and then performed DNase-Seq. We identified 6,805 iDHSs that were at least 2-fold inducible in T cells cultured with IL-2 in each of two replicates (Figs 4A and EV3A), and in parallel, 1,399 DHSs that were decreased by at least 2-fold in T cells stimulated cells in the absence of IL-2 (Figs 4B and EV3B). By intersecting these data sets, we defined 1,015 IL-2-dependent iDHSs (IL-2 iDHSs) (Figs EV3C and 4A–C, Table EV1). Both the entire group of 6,805 iDHSs and the subset of 1,015 IL-2-dependent iDHSs had essentially the same molecular signature, i.e., being highly enriched for motifs for the inducible AP-1, NFAT, and EGR TF families (Figs 4D and EV3D), which is consistent with their function as inducible transcriptional enhancers (Bevington *et al*, 2016). Furthermore, the IL-2

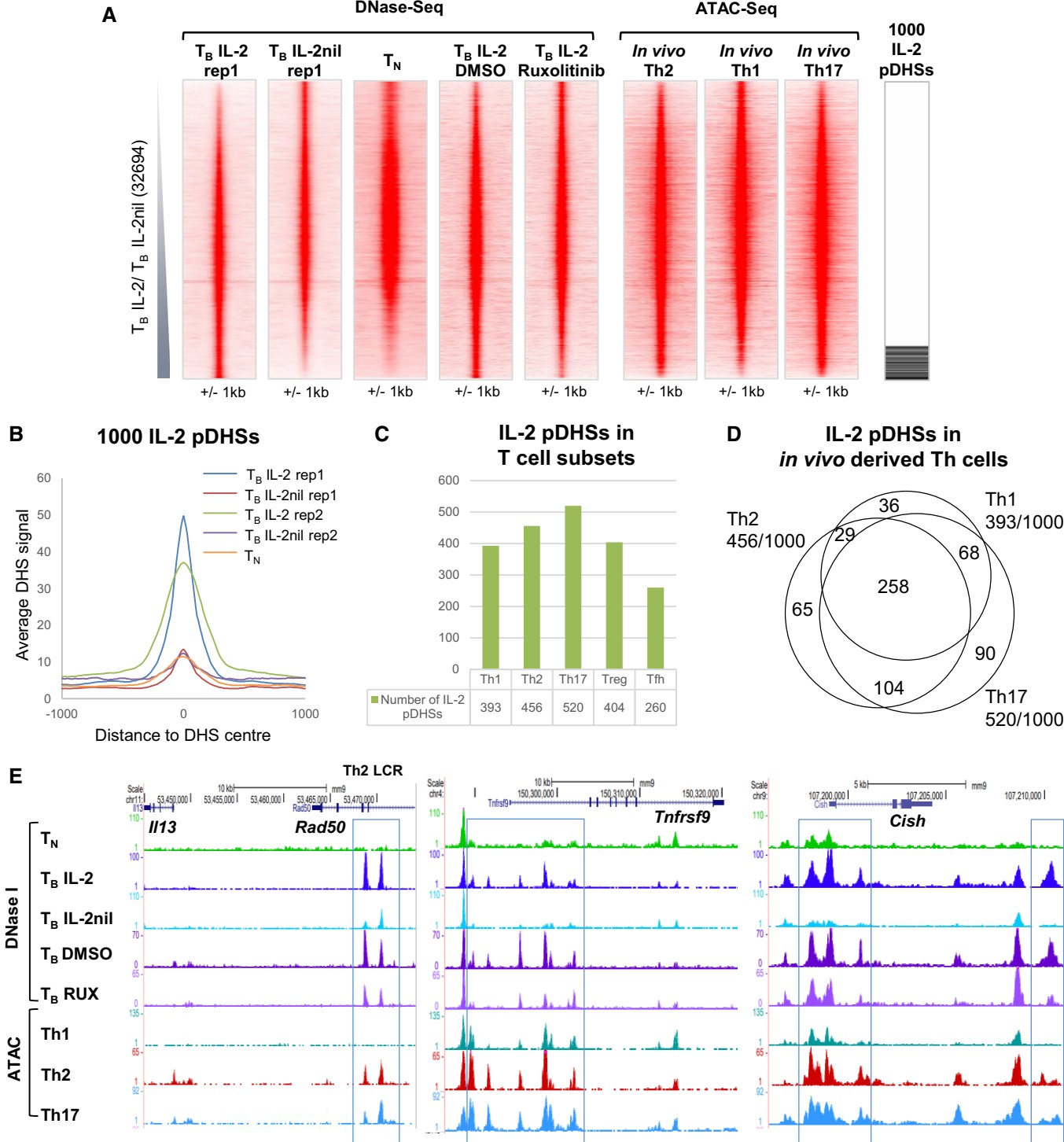

**Figure 2. A subset of pDHSs in previously activated T cells are dependent on IL-2.**

A DNase-Seq tag density plots showing all peaks detected in replicate 1 of T_B IL-2 and T_B IL-2nil (32,694) ordered by increasing fold change of tag count for T_B IL-2 compared to T_B IL-2nil. The sequence tag density is also shown in parallel at the same sites for the DNase-Seq analyses of CD4 naïve T cells (T_N), T_B IL-2 DMSO, and T_B IL-2 Ruxolitinib and ATAC samples from *in vivo*-derived Th2 (Shih *et al*, 2016), Th1, and Th17 cells. The location of the 1,000 IL-2 pDHSs is shown to the right.

B Average DNase-Seq tag density profile for the 1,000 IL-2 pDHSs in both replicates of T_B IL-2 and T_B IL-2nil samples, and for naïve T cells (T_N).

C Number of IL-2 pDHSs which are detected in *in vivo* purified CD4 T cell subsets (Shih *et al*, 2016; Garg *et al*, 2019; Chen *et al*, 2020).

D Overlap between the subsets of IL-2 pDHSs which are present in *in vivo*-derived Th1, Th2, and Th17 cells.

E UCSC genome browser tracks with examples of IL-2 pDHSs within the locus control region (LCR) within the *Rad50* gene which controls the Th2 cytokines *Il13, Il4, and Il5*, plus IL-2 pDHSs within the *Tnfrsf9 and Cish* loci. IL-2 pDHSs are highlighted with blue boxes.

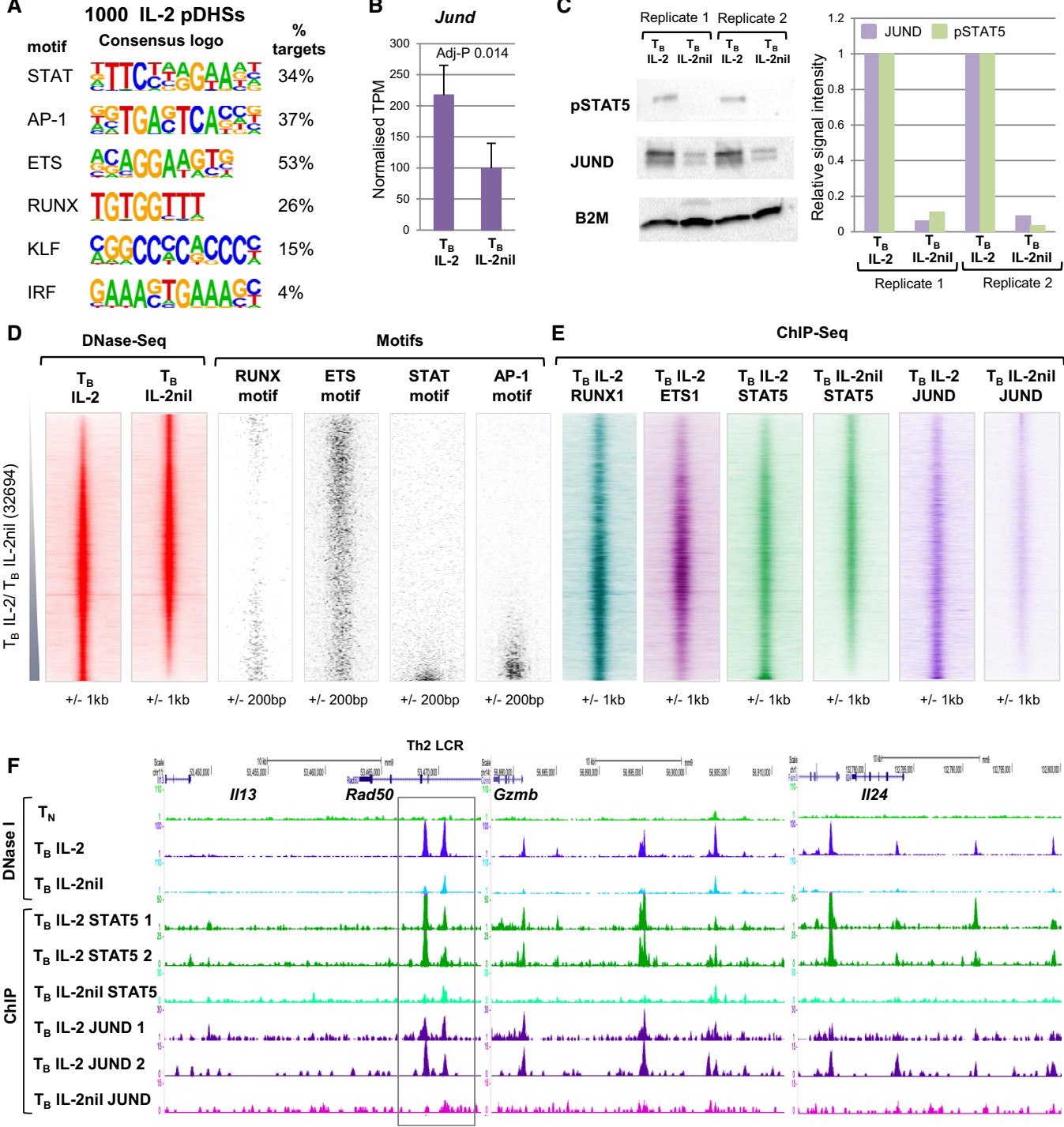

**Figure 3. IL-2 pDHSs bind IL-2 regulated factors.**

A De novo identification of DNA consensus motifs enriched in the 1,000 IL-2 pDHSs determined by HOMER.

B Jund mRNA levels in $T_B$ IL-2 and $T_B$ IL-2nil. Standard deviation is shown from the mean of 3 samples. P-values were calculated from the RNA-Seq data using Limma with Benjamini–Hochberg correction for multiple testing.

C Western blot analyses showing JUND and pSTAT5 protein levels in $T_B$ IL-2 and $T_B$ IL-2nil (left panel) with the signal calculated from 2 replicates relative to $T_B$ IL-2 in the right panel.

D DNA motif density plots (right) showing the location of binding motifs for RUNX, ETS, STAT, and AP-1 transcription factors at the 32,694 DHSs detected in $T_B$ IL-2 and $T_B$ IL-2nil, as depicted in Fig 2A, ordered according to fold enrichment of DNase-Seq tag counts in $T_B$ IL-2 compared to $T_B$ IL-2nil (left).

E DNA sequence tag density plots from ChIP-Seq assays of RUNX1, ETS1, STAT5, and JUND in $T_B$ IL-2, and STAT5 and JUND in $T_B$ IL-2nil ordered as in (D).

F UCSC genome browser tracks at the Il13/Il4/Il5 Th2 cytokine locus LCR (indicated by the gray box within the Rad50 gene), Gzmb and Il24 showing DNase-Seq and ChIP-Seq for STAT5 and JUND in $T_B$ IL-2 and $T_B$ IL-2nil. Replicate JUND and STAT5 ChIP-Seq tracks are shown for $T_B$ IL-2.

iDHSs were not inducible in $T_N$ cells which had not yet undergone reprogramming or been exposed to IL-2 (Fig 4C). Taken together, these data suggest that the γc cytokines as a family have the potential to maintain a long-term transcriptional memory of the gene activation program that is initiated by TCR signaling and autocrine IL-2 expression when T cells become activated for the first time. This program appears to be (i) established when naïve T cells transform into blast cells, (ii) supported by IL-2 during their expansion phase, and IL-4 during Th2 differentiation, and (iii) is likely to be subsequently maintained by the homeostatic cytokines IL-7 and IL-15 when T cells return to the quiescent state as memory T cells. This model of locus priming is further supported by the fact 10% of the 1,000 IL-2-dependent pDHSs had a IL-2-dependent iDHS within 25 kb (Fig 4E), suggesting a functional local cooperation between the two distinct classes of regulatory elements. This close cooperation is consistent with the model present in Fig 1A. Examples of this are shown at the *Csf2/Il3* and *Il10* loci, where both primed and inducible DHSs are present in T-blast cells whereas these loci are devoid of priming in naïve T cells where the inducible sites cannot be formed (Fig 4F). At the *Il10* locus, the same IL-2 pDHSs are known to be to be strongly primed when T cells are rendered tolerant by escalating dose peptide immunotherapy and converted to Tr1-like regulatory T cells (Bevington *et al*, 2020).

## Loss of IL-2-dependent pDHSs correlates with reduced gene expression

To determine the extent by which IL-2 signaling is required to maintain efficient inducible responses in previously activated T cells, we investigated (i) whether the removal of IL-2 had a direct impact on steady-state gene expression, and (ii) whether the associated decrease in pDHS and iDHS formation in the absence of IL-2 affected the level of inducible gene expression. We performed RNA-Seq analysis in $T_B$ cells ± IL-2 ($T_B$ IL-2 and $T_B$ IL-2nil) and ± PMA/I ($T_{B+}$ IL-2 and $T_{B+}$ IL-2nil) (Fig 5A and B). While 51% of the variance in gene expression was accounted for by activation with PMA/I, regardless of whether the cells had been exposed to IL-2 ($T_B$ IL-2 compared to $T_{B+}$ IL-2, and $T_B$ IL-2nil compared to $T_{B+}$ IL-2nil), 22% of the variance was attributed to the presence of IL-2 (Fig 5A and B). Furthermore, correlation clustering showed that the presence of IL-2 had the biggest impact on gene expression prior to stimulation ($T_B$ IL-2 compared to $T_B$ IL-2nil; Fig 5B) with 612 genes being more than 3-fold upregulated in $T_B$ IL-2 compared to $T_B$ IL-2nil and 356 genes that were at least 3-fold down-regulated when IL-2 was present (adj $P < 0.05$) (Fig 5C, Table EV2). Many of the IL-2-dependent genes were associated with the cell cycle, such as the histone genes and the cell division cycle (cdc) genes (Table EV2), consistent with a role for IL-2 in this process. Genes which became more active when IL-2 was removed included genes involved in T cell homeostasis which are normally associated with a cell signature more reminiscent of naïve T cells, such as *Tcf7*, *Ccr7*, and *Cd62l* (*Sell*).

To examine the impact of IL-2 pDHSs on IL-2-dependent gene expression, we plotted the fold change in gene expression between $T_B$ IL-2nil compared to $T_B$ IL-2 for the nearest gene to all the DHSs plotted in the union of $T_B$ IL-2 and $T_B$ IL-2nil (Fig 5D). Genes associated with IL-2 pDHSs showed a general trend of being dependent upon IL-2 (Fig 5D). These included immune regulatory genes such as *Tnfrsf9* and *Cish* (Fig 2E), *Gzmb* (Fig 3F), and *Socs1* (Fig 5E). At

the *Sell* locus, RNA expression is higher in $T_B$ IL-2nil consistent with the formation of DHSs which are also present in naïve cells (Fig 5F), a trend which is generally observed at the IL-2nil upregulated DHSs (Fig 5D). Overall, these data indicate that primed DHSs which are either created or erased in the presence of IL-2 do play a role in controlling IL-2-dependent changes in steady-state gene expression in previously activated T cells.

## IL-2 signaling is required for rapid reactivation of some key immune response genes

The above correlation clustering indicated that the presence of IL-2 also modulated the inducible gene repertoire of recently activated T cells, but to a much lesser extent than seen for steady-state gene expression (Fig 5B). To further investigate this subset, we identified 465 genes which were upregulated at least 3-fold in $T_B+$ compared to $T_B$ (Table EV2) and assessed whether recent prolonged exposure to IL-2 was required for efficient and rapid reactivation of these genes. However, removal of IL-2 in $T_B+$ IL-2nil cells affected the induction of just 14 genes by at least twofold (Table EV2). While the number of genes was modest, it is striking that many key immune response genes and genes harboring IL-2-dependent pDHSs and iDHSs were included in this group, such as *Il13, Ifng, Csf2, Il3, Il24,* and *Il31*. For some of these genes, the data were further validated by quantitative PCR analysis (Fig 6A). We also demonstrated that the same genes were less inducible when $T_B+$ IL-2 cells were treated with the JAK1/2 inhibitor Ruxolitinib which inhibits signaling from IL-2 to STAT5 (Fig 6B). The dependence upon IL-2 signaling was exemplified at the *Il3/Csf2* and *Il13* loci where withdrawal of IL-2 led to a decrease in inducible gene expression that was linked to a decrease in both primed and inducible DHSs (Fig 6C).

## The role of IL-2 in maintaining chromatin priming can be replaced by IL-7

The above studies established that IL-2 maintains chromatin priming at IL-2 pDHSs in $T_B$ cells cultured *in vitro* (Fig 2A) and that this form of priming exists in memory T cells formed *in vivo* (Fig 2C–E). Although IL-2 is likely to contribute to the initial reprogramming of naïve T cells to effector T cells *in vivo*, the long-term maintenance of IL-2 pDHSs most likely relies on alternative homeostatic cytokines such as IL-7 in lymph nodes, or IL-15 in tissues, which also activate γc family receptors (Hara *et al*, 2012; Raeber *et al*, 2018).

To investigate whether other γc receptors can fulfill the same role as IL-2, we began by replacing IL-2 with IL-7 in the $T_B$ culture model employed above (Fig 1C). Duplicate DNase-Seq analyses confirmed that IL-7 had the ability to reproducibly maintain a similar subset of pDHSs as compared to those maintained by IL-2 and that these pDHSs were also lost upon IL-7 withdrawal (Fig 7A), as depicted by the average sequence tag density at the 1,000 sites (Fig 7B). As shown above for IL-2, the IL-7-dependent pDHSs included sites in the Th2 LCR within *Rad50*, and sites at the *Tnfrsf9* and *Cish* loci (Fig 7C), while sites at the CD3 gene cluster and *Tbp* loci were unaffected by IL-7 (Fig 7D). In support of the data in Fig 4C for IL-2, we also demonstrated that a subset of iDHSs were less inducible in $T_B+$ IL-7 cells after IL-7 withdrawal (Fig 7E) and that this affected the expression of inducible genes in $T_{B+}$ IL-7nil cells compared to $T_{B+}$ IL-7 cells (Fig 7F).

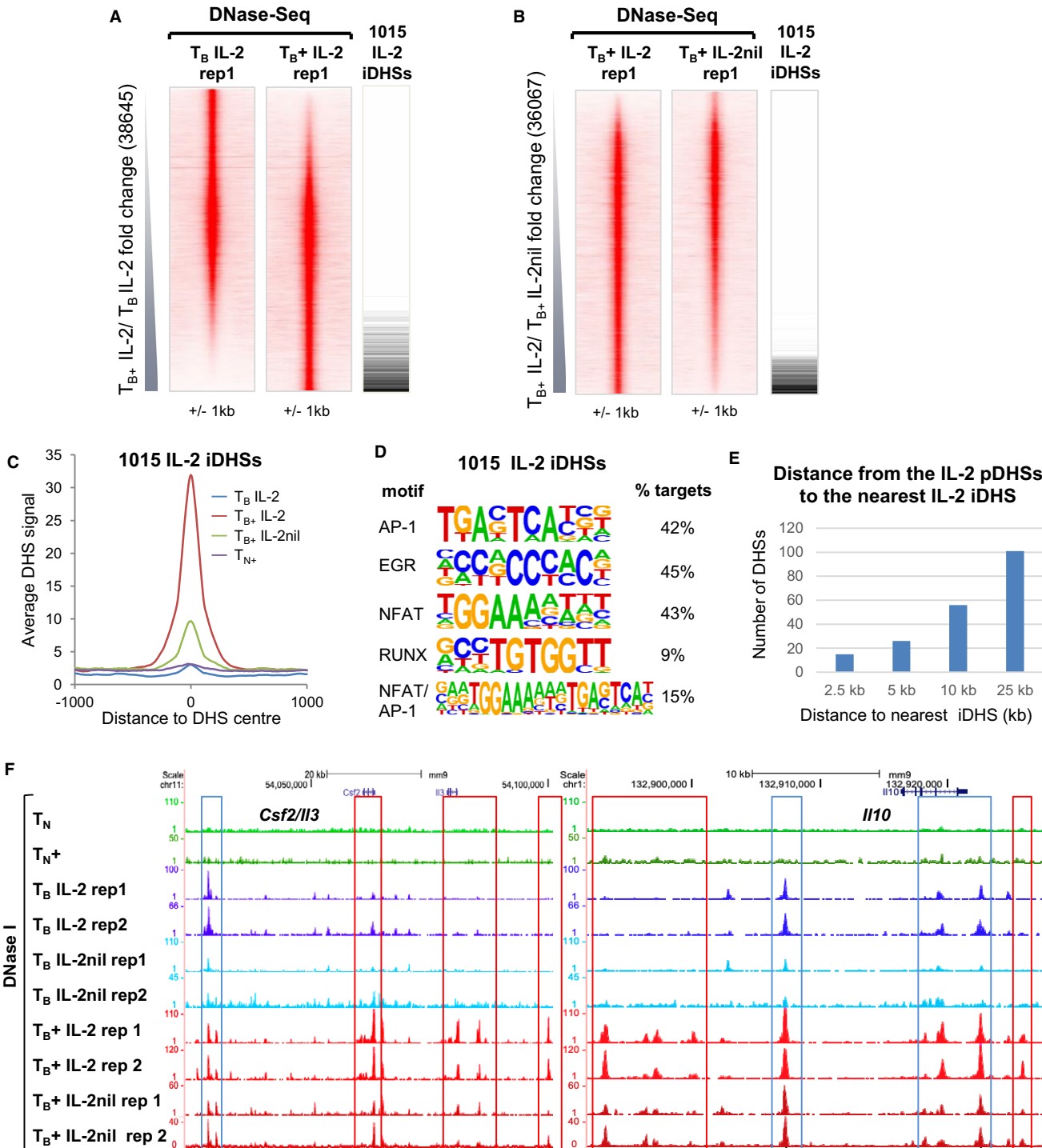

**Figure 4. IL-2 signaling supports PMA/I-inducible DHS formation.**

A, B DNase-Seq tag density plots showing all peaks detected in replicate 1 of $T_B$ IL-2 and $T_{B+}$ IL-2 (A, 38,645) and in $T_{B+}$ IL-2 and $T_{B+}$ IL-2nil (B, 36,067), ordered by increasing fold change in tag count as indicated. The position of the 1,015 IL-2-dependent inducible DHSs is shown alongside.

C Average DHS signal at the 1,015 IL-2-dependent inducible DHSs in $T_B$ IL-2, $T_{B+}$ IL-2, $T_{B+}$ IL-2nil, and naïve T cells stimulated with PMA/I ($T_{N+}$).

D De novo identification of DNA consensus motifs enriched at the 1,015 IL-2 iDHSs as determined by HOMER.

E Bar graph showing the number of IL-2 iDHSs found within 25 kb of an IL2 pDHS.

F Examples of IL-2 iDHSs which are proximal to IL-2 pDHSs at the Csf2/Il2 and Il10 loci. IL-2 iDHSs are highlighted in red and IL-2 pDHSs in blue.

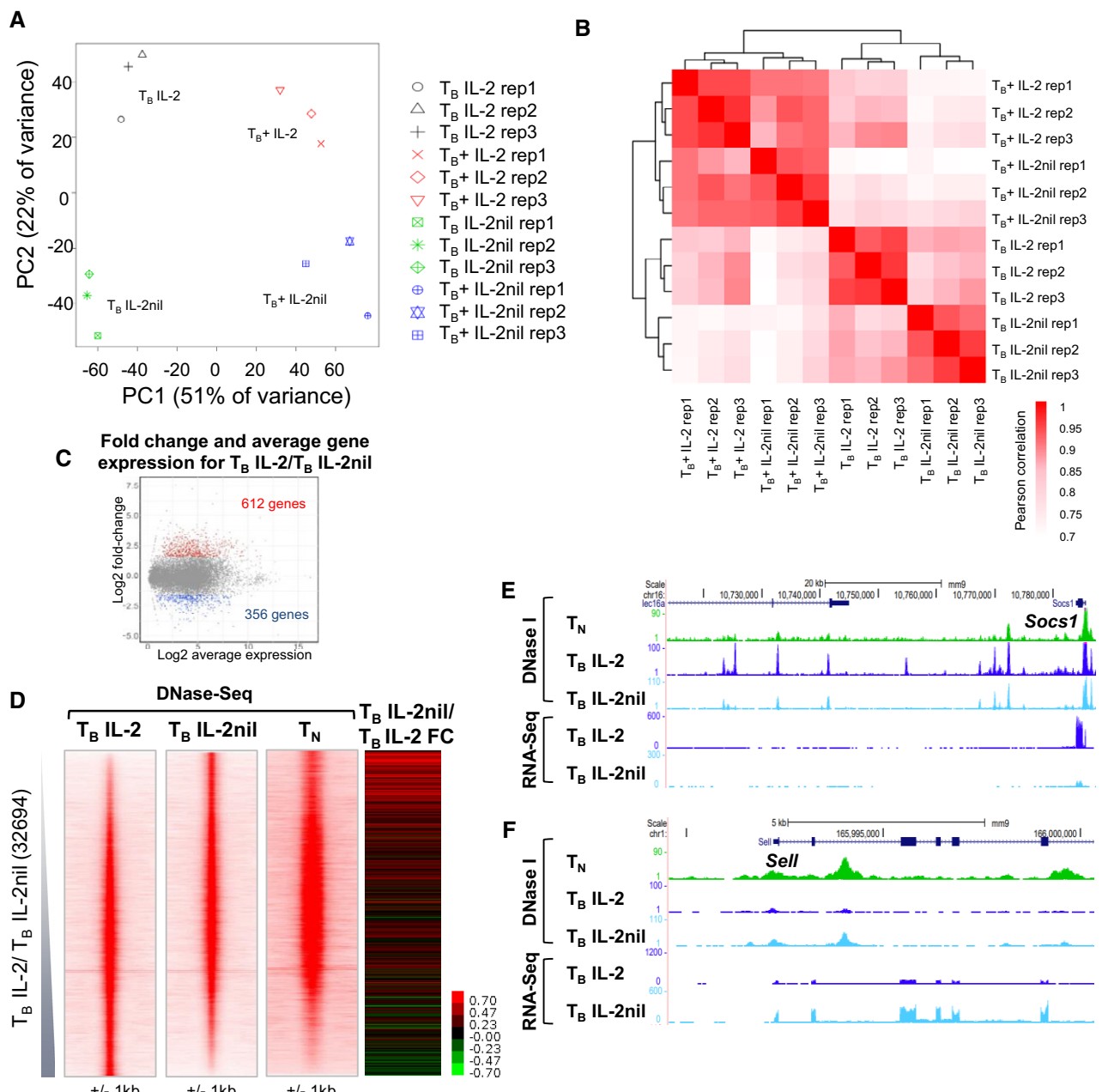

**Figure 5. Loss of IL-2 priming correlates with reduced gene expression of proximal genes.**

A   Principle component analysis (PCA) of $T_B$ IL-2, $T_B$ IL-2nil, $T_B+$ IL-2, and $T_B+$ IL-2nil from 3 replicates.

B   Hierarchical clustering of RNA-Seq data from 3 replicates as described in (A).

C   MA plot showing the genes differentially regulated between $T_B$ IL-2nil and $T_B$ IL-2. The number of genes with a fold change > 3 and adj *P*-value < 0.05 are highlighted. *P*-values were calculated from the RNA-Seq data using Limma with Benjamini-Hochberg correction for multiple testing.

D   $\log_2$ fold change in gene expression in $T_B$ IL-2nil compared to $T_B$ IL-2 for the nearest gene to the DHSs ranked according to the fold change in DNase-Seq tag count for $T_B$ IL-2/$T_B$ IL-2nil. The DNase-Seq tag density plots are taken from Fig 2A for $T_B$ IL-2, $T_B$ IL-2nil, and $T_N$.

E, F   UCSC genome browser tracks showing DNase-seq and RNA-Seq at the *Socs1* (E) and *Sell* loci (F).

## The IL-7 receptor is required to maintain a subset of IL-2 pDHSs in Ag-specific memory T cells

The detection of 39% of the IL-2 pDHSs in *in vivo*-generated 2W1S-specific Th1 memory T cells (Fig 2C) 56 days post-infection

indicates that these sites are epigenetically stable when the TCR signals are down-regulated. We therefore investigated whether the stable *in vivo* maintenance of pDHSs was IL-7-dependent using $CD4^{\text{creERT2}} \times Il7r^{\text{f/f}}$ to inducibly delete IL-7Rα expression once Ag-specific memory CD4 T cells had formed (Fig 8A). These mice were

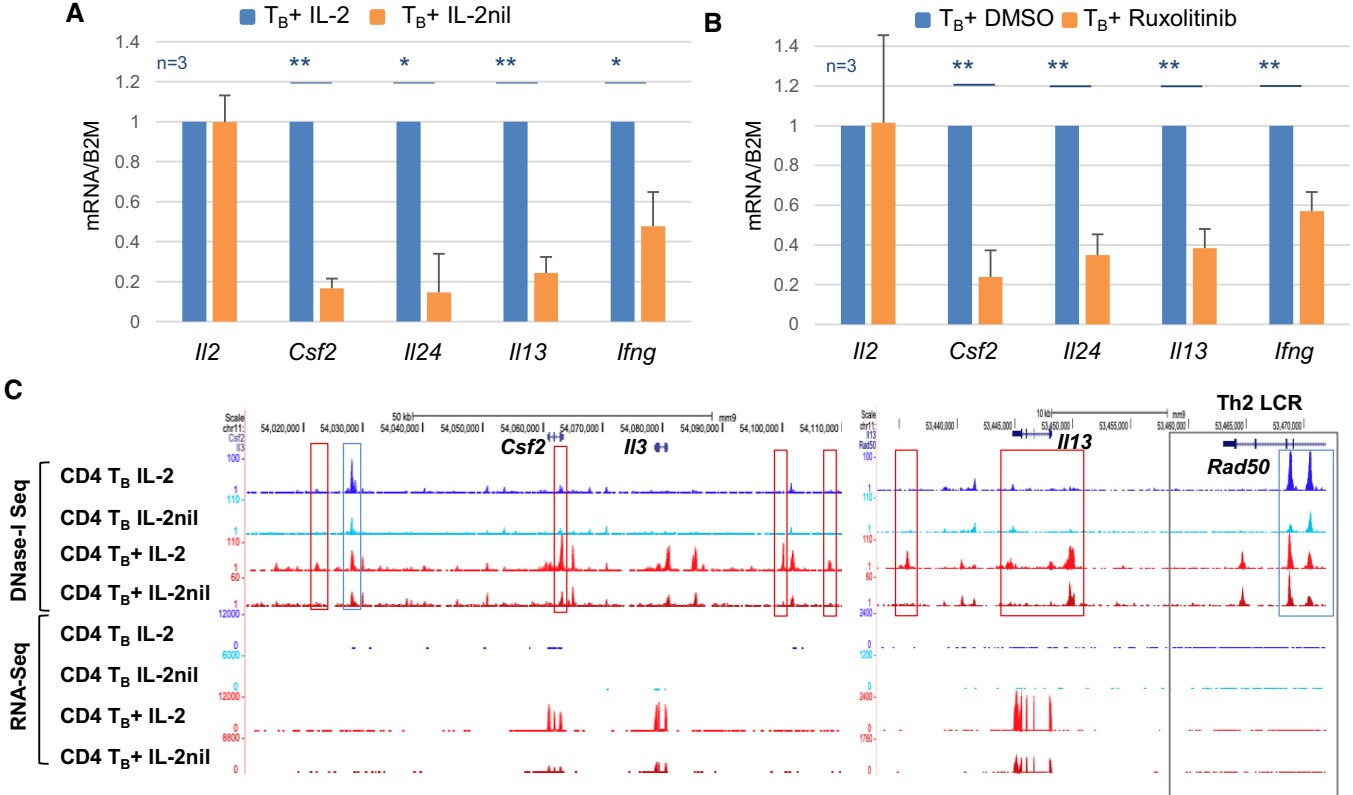

**Figure 6. Loss of IL-2 priming correlates with reduced gene expression of some inducible genes.**

A, B Relative mRNA expression levels in $T_B+$ IL-2 and $T_B+$ IL-2nil (A) and $T_B+$ DMSO and $T_B+$ Ruxolitinib (B). The expression level in the IL-2/DMSO $T_B+$ sample is set to 1. ($n = 3$, error bars represent standard deviation, $P$-values were calculated using a paired $t$-test with $*P < 0.05$, $**P < 0.005$, $***P < 0.0005$.)

C  UCSC genome browser tracks showing DNase-Seq and RNA-Seq at the IL-2 regulated genes *Csf2/Il3* and at the LCR within the *Rad50* gene (gray box) which controls the Th2 cytokines *Il13, Il4, and Il5*. IL-2 pDHSs are highlighted in blue and IL-2 iDHSs in red.

further crossed onto a *ROSA26*[tdRFP] background to facilitate detection of cre expression. *CD4*[creERT2] × *Il7r*[f/f] mice and their *Il7r*[f/f] mice littermate controls were infected with *Lm-2WIS* to generate Th1 cells as in Fig 1D. After approximately 3 weeks, administration of tamoxifen was used to induce cre-mediated excision of exon 3 of the *Il7r* gene (Fig 8A). Flow cytometric analyses 2 weeks later confirmed efficient deletion of IL-7Rα expression on the 2W1S-specific CD4 memory T cells (Fig 8B). To determine the impact of loss of IL-7Rα expression by memory CD4 T cells, 2W1S:I-A[b+],CD44[+], RFP[+], IL-7Rα[−] CD4 T cells were sorted from *CD4*[creERT2] × *Il7r*[f/f] mice, alongside 2W1S:I-A[b+], CD44[+], IL-7Rα[+] CD4 T cells from *Il7r*[f/f] controls and harvested for ATAC-Seq. The number of IL-7rα-deficient cells recovered from the *CD4*[creERT2] × *Il7r*[f/f] mice was 40% of the number of IL7rα+ cells harvested from the *Il7r*[f/f] controls (Fig 8C), suggesting that loss of IL-7Rα leads to a 2.5-fold reduction in Th1 memory T cell expansion and/or survival.

A substantial reduction in ATAC-Seq signal after disruption of *Il7r* was apparent at the specific subset of 411 of the 1,000 IL-2 pDHSs which contain a STAT-binding motif (Figs 8D and EV4A). This reduction, while relatively modest, is consistent across three replicates and likely reflects the reality that signals through other gamma chain receptors such as IL-2, IL-4, and IL-15 are still active. Examples can be seen at the *Gzmb, Il4ra,* and the *Pilrb* loci (Fig 8E)

where the IL-2 pDHSs are reduced in samples where the IL-7 receptor is inactivated. Furthermore, these sites bind STAT5 in $T_B$ IL-2 and this binding is reduced when the IL-2 is absent. Control regions at the *Tbp* locus and the CD3 gene cluster are included to show that a general loss of signal is not observed (Fig EV4B).

**IL-2 pDHSs play a role in terminal T cell differentiation**

The relatively modest effect of IL-2 withdrawal on inducible gene expression in recently activated T cells (Fig 6) led us to look for additional roles for IL-2 dependent priming later in T cell development. The transformation of naïve T cells to actively dividing Th0 blast cells represents just the first stage of T cells developing further into various alternative types of mature differentiated effector T cells equipped for combatting different types of infections (Fig 1B). The different branches of terminal T cell differentiation involve additional alternative cytokine and TF-dependent epigenetic reprogramming events subsequent to the TCR/CD28-dependent reprogramming that occurs during blast cell transformation to Th0 T cells (Fig 1B). We demonstrated above that many of the IL-2-dependent pDHSs are hypersensitive in primary Th cells (Figs 2A and 2C–E) and that a Th2-specific cytokine gene locus harbors primed sites which are sensitive to loss of IL-2 signals (*Il13*; Fig 3F).

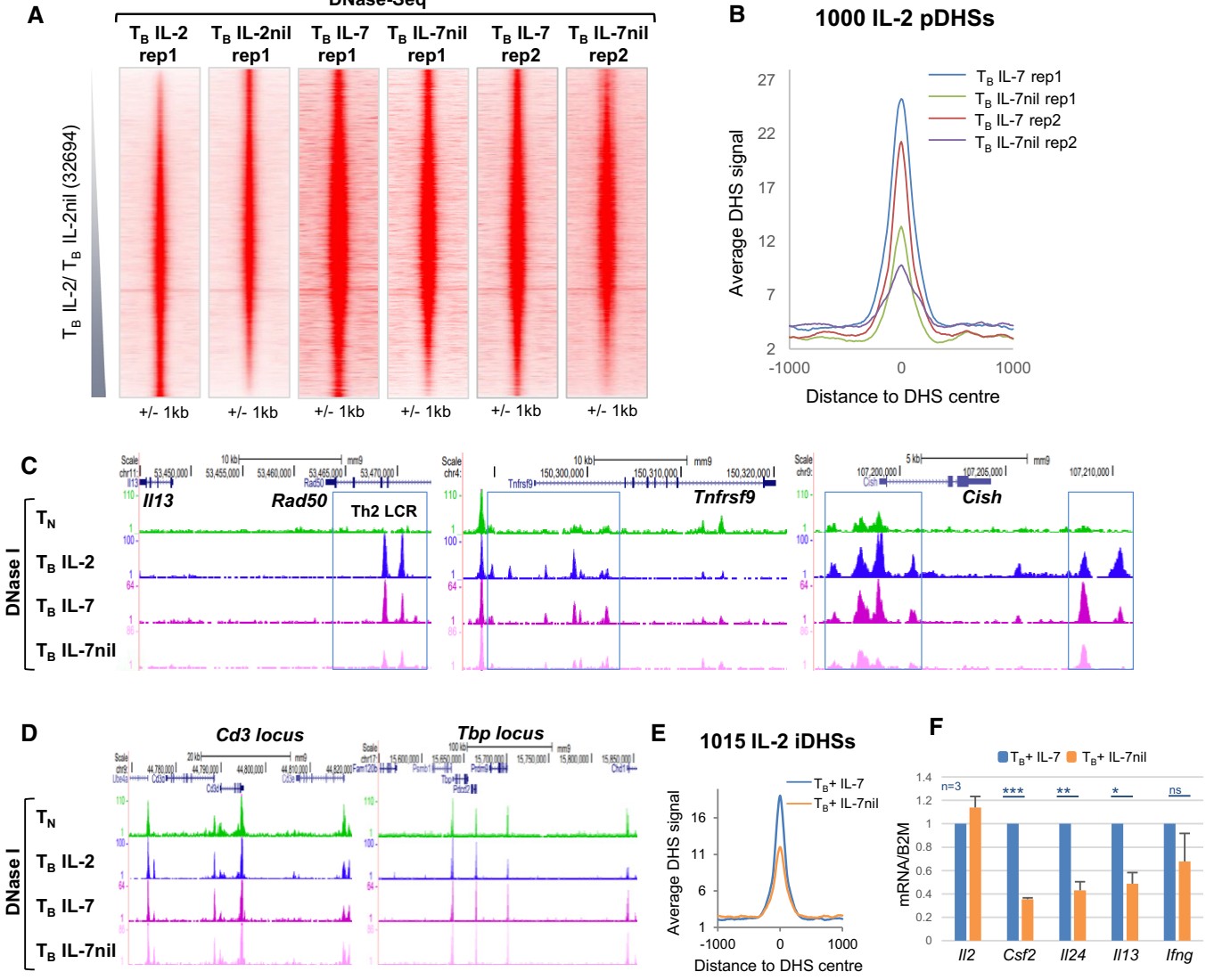

**Figure 7. IL-2 pDHSs can be maintained *in vitro* by IL-7 in place of IL-2.**

A   DNase-Seq tag density plots showing all peaks detected in replicate 1 of T_B IL-2 and T_B IL-2nil (data taken from Fig 2A), duplicate DNase-seq samples for T_B IL-7 and T_B IL-7nil are shown in parallel for the same sites.

B   Average DNase-Seq tag density profile for the 1,000 IL-2 pDHSs in both replicates of T_B IL-7 and T_B IL-7nil.

C, D   UCSC genome browser tracks with DNase-Seq data for T_B IL-7 and T_B IL-7nil showing examples of IL-2 pDHSs within the locus control region (LCR) within the *Rad50* gene which controls the Th2 cytokines *Il13, Il4, and Il5*, plus IL-2 pDHSs within the *Tnfrsf9 and Cish* loci (C), and non-primed control regions from the CD3 gene cluster and the *Tbp* locus (D). IL-2 pDHSs are highlighted with blue boxes.

E   Average DHS signal at the 1,015 IL-2 regulated inducible DHSs in T_B+ IL-7 and T_B+ IL-7nil.

F   Relative mRNA expression levels in T_B+ IL-7 and T_B+ IL-7nil. The expression level in the T_B+ IL-7 is set to 1. (*n* = 3, error bars represent standard deviation, *P*-values were calculated using a paired *t*-test with **P* < 0.05, ***P* < 0.005, ****P* < 0.0005.)

To further explore the interplay between IL-2 pDHSs, lineage-determining TFs, and specific STAT proteins, we analyzed publically available data sets from terminally differentiated T cells. We intersected STAT4 and T-BET ChIP-Seq peaks from Th1 cells (Wei *et al*, 2010; Gokmen *et al*, 2013), STAT6 and GATA3 peaks from Th2 cells (Wei *et al*, 2010, 2011), and STAT3 and RORγt from Th17 cells (Durant *et al*, 2010; Ciofani *et al*, 2012) and overlapped these with the 1,000 IL-2 pDHSs. Strikingly, we found that 62.5% of the IL-2 pDHSs go on to bind both an alternative cytokine-specific STAT

protein and a lineage-determining TF in Th1, Th2, or Th17 cells (Fig 9A). As an additional control for these studies, we also used a parallel method of inducing T cell blast transformation using CD3 and CD28 antibodies in place of ConA. This analysis confirmed that the same set of DHSs (82–89%) were also primed by IL-2 in T-blast cells generated by a more physiological method, which rules out the possibility of inappropriate priming by ConA (Fig 9A). These observations are summarized by the model in Fig 9B which illustrates priming of gene regulatory elements by AP-1 and STAT5 prior to

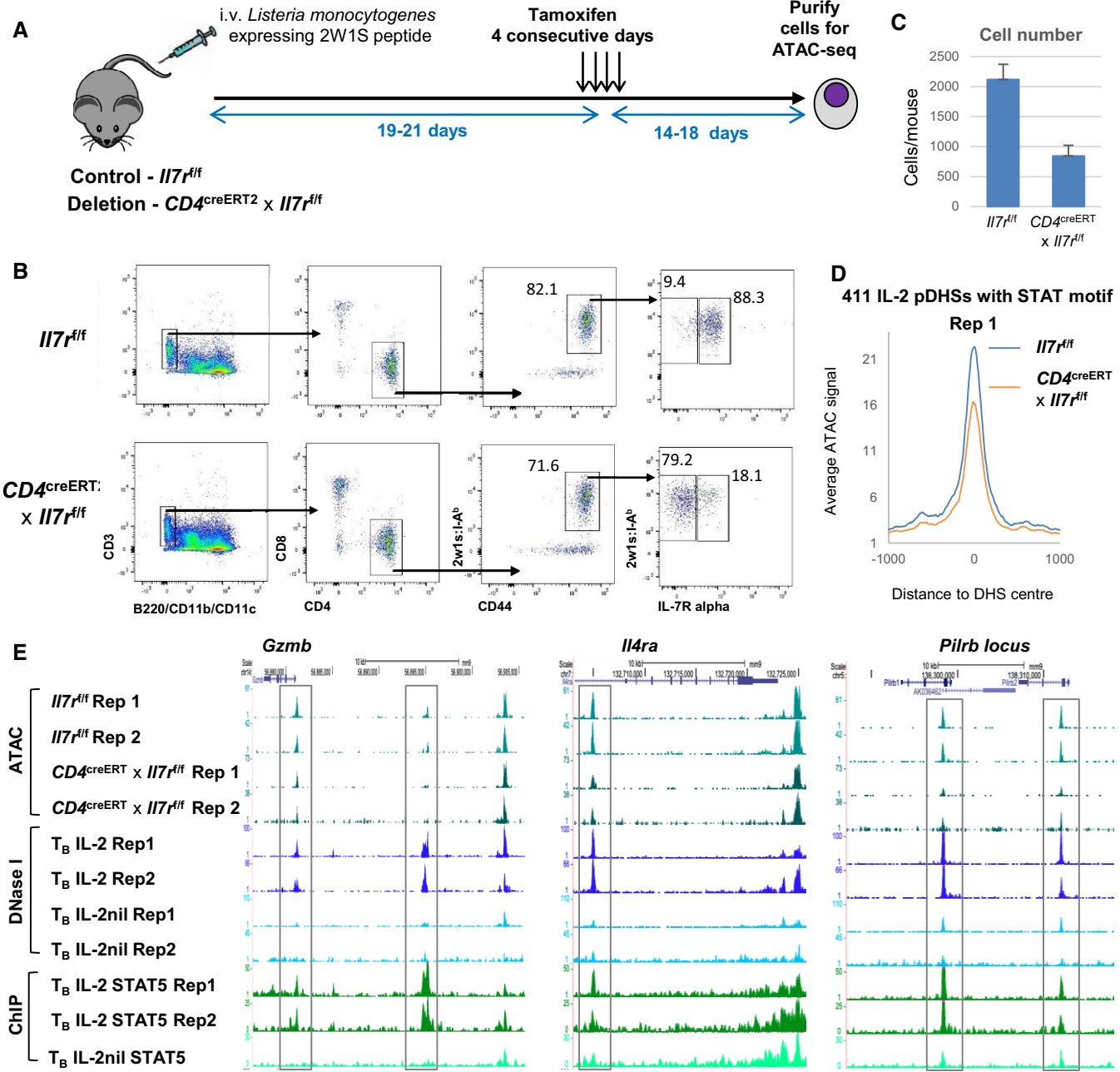

**Figure 8.  IL-2 pDHSs are reduced in antigen-specific memory T cells upon deletion the IL-7 receptor.**

A  Experimental overview of the tamoxifen-inducible disruption of *Il7r* following generation of 2W1S-specific Th1 memory T cells. *CD4*^creERT2^ × *Il7r*^f/f^ or *Il7r*^f/f^ (littermate control) mice were injected i.v. with *Lm*-2W1S, tamoxifen administered for 4 consecutive days (19–21 dpi), and spleen cells harvested for ATAC-Seq 14–18 days later.

B  Flow cytometric analysis of IL-7Rα expression by 2W1S-specific CD44^hi^ CD4 T cells in the presence or absence of cre recombinase.

C  Average number of 2W1S-specific cells recovered/mouse after sorting 3 replicate samples of *Il7r*^f/f^ compared to *CD4*^creERT2^ × *Il7r*^f/f^. Errors bars indicate the standard deviation.

D  Average ATAC-Seq signal at the 411 IL-2 pDHSs which contain a STAT-binding motif.

E  UCSC genome browser tracks at loci with IL-2 pDHSs showing replicate ATAC-Seq samples for *CD4*^creERT2^ × *Il7r*^f/f^ memory T cells, *Il7r*^f/f^ control memory T cells, replicate DNase-Seq samples for $T_B$ IL-2 and $T_B$ IL-2nil and $T_B$ IL-2 and $T_B$ IL-2nil STAT5 ChIP-Seq samples. The gray boxes signify IL-2 pDHSs which diminish after disruption of the *Il7r* gene.

exposure to differentiation-inducing signals. This model is exemplified by UCSC browser screenshots depicting key genes representative for the different T cell subsets: The *Tbx21* and *Il12rb2* loci in

Th1 cells, the Th2 cytokine locus (*Il4* and *Il13*) in Th2 cells, and the *IL17a* and *Il12rb2* loci in Th17 cells (Fig 9C). In each case, the DHSs which bind lineage-specific STAT proteins and TFs also bind STAT5

in the T-blast (Th0) cells prior to differentiation. This suggests a bookmarking mechanism, whereby IL-2/IL-7-inducible STAT5 and/or AP-1 maintain the pDHSs prior to differentiation. This renders $T_B$ cells permissive to responding to alternative gene regulation programs when the STAT proteins are exchanged or complemented by lineage-determining factors binding to the same accessible regions. Hence, priming by IL-2 or IL-7 responsive TFs is crucial for keeping genes accessible and receptive to fate-determining factors encountered later. Interestingly, although GATA3 and STAT6 are not required for the initial opening up of the first two DHSs that appear within the Th2 LCR in Th0 cells, two additional sites within the LCR recruit GATA3 downstream, and STAT6 upstream of the two central sites which bind GATA3 and STAT6 after differentiation to Th2 cells (Fig 9C). Similar phenomena are observed at the *Il17a* and *Il12rb2* loci. This suggests a model, whereby Th lineage-specific genes are activated in a stepwise process whereby the initial priming is driven by IL-2 in Th0 cells and the line-specific program is consolidated by other lineage-defining cytokines and transcription factors which can also activate additional regulatory elements.

**Many IL-2 pDHSs are shared between Th lineages**

It is significant that a large proportion of IL-2-dependent pDHSs are shared between Th1, Th2, and Th17 cells. However, this also raises questions as to the various roles of IL-2 because IL-2 was previously thought to play opposing roles during the differentiation of Th1 and Th17 cells. While IL-2 promotes the formation of Th1 cells, it inhibits the differentiation of the Th17 lineage (Laurence *et al*, 2007; Liao *et al*, 2011b; Yang *et al*, 2011; Fujimura *et al*, 2013) which are also dependent on the AP-1 family member JUNB. *Junb* expression increases during polarization toward the Th17 lineage and conditional deletion of *Junb* impairs the expression of classical Th17 genes (Carr *et al*, 2017). While we observe PMA/I-inducible *Junb* expression in undifferentiated T-blast cells (Fig 10A), it is unlikely to play a role in maintaining the IL-2 pDHSs as, unlike JUND, it does not bind to the pDHSs in the absence of stimulation with PMA/I (Fig 10B; Bevington *et al*, 2016), and the inducible levels of *Junb* mRNA are actually greater in the absence of IL-2 (Fig 10A). Consistent with the IL-2 pDHSs being maintained in differentiated T cells, published data show that 41% of the IL-2 pDHSs are bound by JUNB in Th17 cells (Carr *et al*, 2017) and over half of these are bound by JUND in $T_B$ IL-2 (Fig 10C). This suggests that the balance of AP-1 factors at the same site could be key to determining Th cell fate when the cells are exposed to different cytokines during polarization. Furthermore, Th17 cell differentiation may be suppressed by IL-2 via the maintenance of JUND binding at the pDHSs. Consistent with this, using published data from Th17 cells where *Junb* is conditionally deleted, we can show that JUND binding to the IL-2 pDHSs increases after loss of JUNB (Carr *et al*, 2017; Fig 10D). This is exemplified at the *Il17a* Th17 cytokine locus where JUND binds to a DHS which is reduced when IL-2 is absent in T-blast cells (Th0) and JUNB binds to the same site in Th17 cells (Fig 10E). Furthermore, when JUNB is ablated, JUND binding increases and, as shown by Carr *et al* (2017), *Il17a* expression is reduced (Fig 10E). This suggests that JUND could play a role in repressing *Il17a* in this context. Indeed, JUND has been implicated as a repressive factor previously (Meixner *et al*, 2004), and in $T_{B+}$ IL-2 cells from *Jund*$^{-/-}$ mice, the inducible genes *Ccl1*, *Csf2*, and *Il24* are boosted compared to $T_{B+}$ cells from *Jund*$^{+/+}$ mice

(Fig 10F). However, although this data suggest that the IL-2-induced repression of Th17 differentiation may be in part mediated by JUND binding to the pDHSs, this is likely to be just one of many mechanisms by which IL-2 suppresses Th17 differentiation. IL-2 is also known to suppress pathways promoting Th17 differentiation by (i) suppressing expression of IL6Rα (Liao *et al*, 2011b) and (ii) by inducing STAT5 which displaces STAT3, leading to loss of activation of Th17 target genes (Yang *et al*, 2011).

Conversely, the binding of JUNB to some of the IL-2 pDHSs represses specific genes in Th17 cells (Carr *et al*, 2017). At the *Irf8* locus, a known repressor of Th17 differentiation (Ouyang *et al*, 2011), the −21 kb IL-2 pDHS is bound by JUND in T-blast cells and JUNB in Th17 cells (Fig 10G; Carr *et al*, 2017). Consistent with this, the subsequent deletion of *Junb* increased JUND binding and *Irf8* expression (Fig 10G; Carr *et al*, 2017). In conclusion, IL-2 pDHSs may display different activity in different Th subsets. This is further reflected by the level of p300 binding which is elevated in Th17 cells and not Th1 cells at the *Il17* locus (Fig 10E) and in Th1 cells only at the *Irf8* locus (Fig 10G; Ciofani *et al*, 2012; Vahedi *et al*, 2012). Therefore, the maintenance of these sites, albeit in a repressed state, may be key to the plasticity between the subsets.

# Discussion

It is well established that cytokine signaling to STAT proteins promotes the activation of lineage-determining TF genes, and in activating lineage-specific gene expression programs in differentiated T cells (Fig 1B; Liao *et al*, 2011a; Vahedi *et al*, 2012; Ross & Cantrell, 2018; Leonard *et al*, 2019). However, it was not previously appreciated just how extensive a role that IL-2 signaling to AP-1 and STAT5 played in making recently activated T cells more receptive to T cell differentiation driven by other STAT proteins such as STAT3, STAT4, and STAT6. The role of STAT5 in differentiation was largely attributed to key regulators such as the IL-4 and IL-12 receptors and their downstream targets (Liao *et al*, 2008, 2011b).

Based on the above data, and previously published studies, we propose that the development and maintenance of the effector T cell repertoire relies on both (i) the initial autocrine IL-2 signaling induced when $T_N$ cells first encounter antigen-presenting cells, and (ii) signaling from other γc family cytokines such as IL-7 and IL-15 once recently activated T cells or memory T cells relocate into lymph nodes or peripheral tissues. The evidence presented here strongly supports a model whereby γc cytokine signaling establishes and/or maintains an active accessible chromatin environment permissive to (i) receiving signals from TFs driving lineage-specific gene expression, and (ii) reactivation by TCR signaling when previously activated T cells re-encounter Ag (Fig 9B). For example, the intensively studied Th2 LCR within the *Rad50* locus comprises seven distinct DHSs in Th2 cells where it controls activation of the Th2-specific IL-4, IL-13, and IL-5 genes (Fields *et al*, 2004). This locus is epigenetically reprogrammed and primed with active histone modifications within 24–48 h of activation of TCR signaling when $T_N$ cells transform into $T_B$ cells (Bevington *et al*, 2016, 2017b). However, just two of these DHSs are initially opened up as epigenetically primed DHSs when Th0 cells are first activated (Bevington *et al*, 2016, 2017b). Significantly, both of these pDHSs, plus two flanking DHSs which are acquired in Th2 cells (Fields *et al*,

2004), have the ability to bind GATA3 and STAT6 in Th2 cells. The first two *Rad50* pDHSs lie directly adjacent to an iDHS (Bevington *et al*, 2016) with known enhancer function (Fields *et al*, 2004)

which suggests that they play a pioneering role in the activation of this locus prior to Th2 differentiation and locus activation. Lineage-determining factors such as T-BET, GATA3, and RORγt may

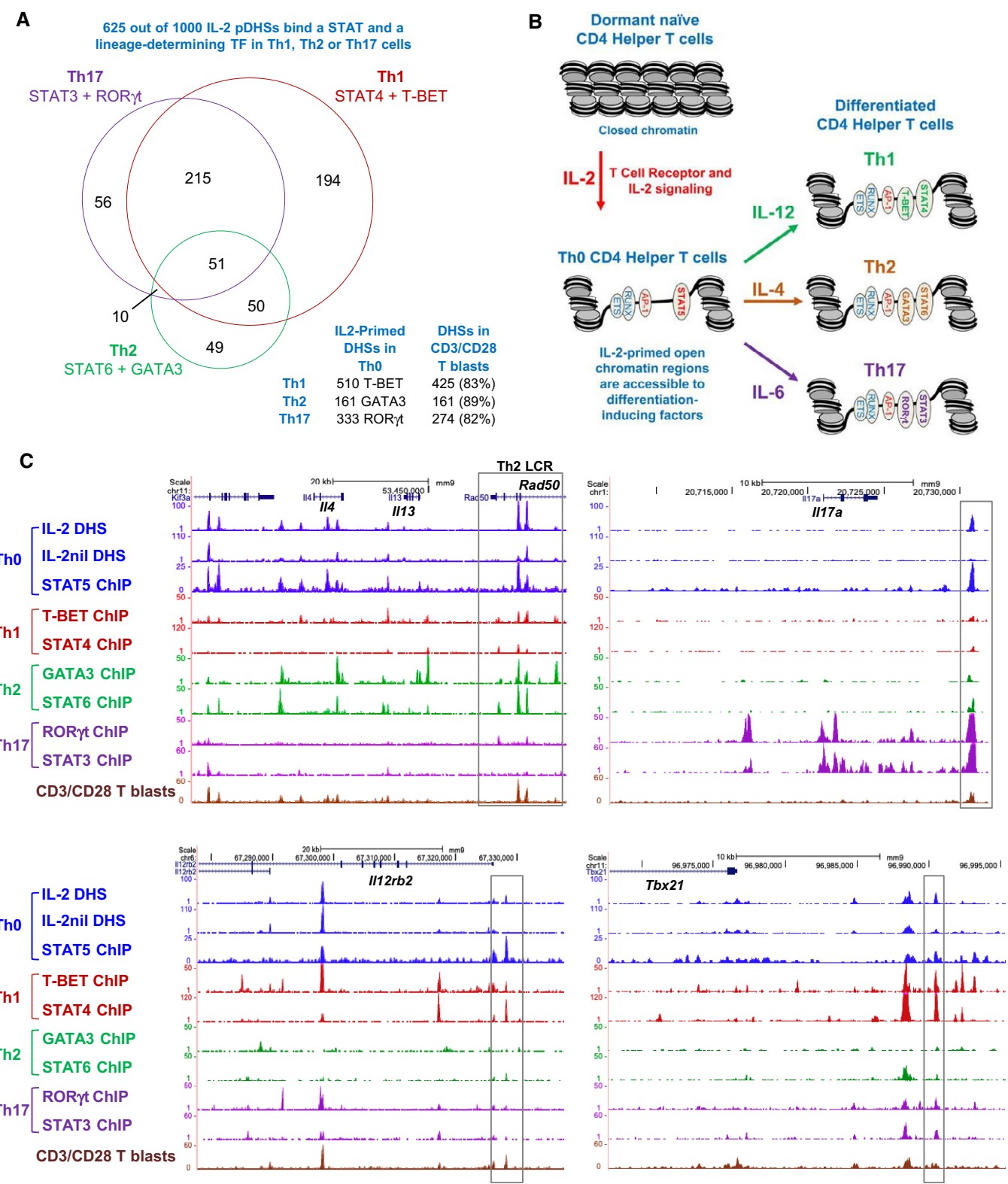

**Figure 9.**

**Figure 9. IL-2 pDHSs are present in Th1, Th2, and Th17 cells.**

A  Venn diagram showing the overlaps between subsets of the 1,000 IL-2 pDHSs which are also identified as ChIP-Seq peaks for STAT6 (Wei *et al*, 2010) and GATA3 (Wei *et al*, 2011) in Th2 cells (161), STAT4 (Wei *et al*, 2010), and T-BET (Gokmen *et al*, 2013) in Th1 cells (510), and STAT3 (Durant *et al*, 2010) and RORγt (Ciofani *et al*, 2012) in Th17 cells (333). The table below indicates the proportion of these peaks also seen in T blasts generated using CD3/CD28 antibodies instead of ConA.
B  Model demonstrating priming of Th0 cells by IL-2-inducible AP-1 and STAT5 prior to binding of lineage-determining factors to the same regions.
C  UCSC genome browser tracks showing ChIP-Seq and DNase-Seq at the LCR within the *Rad50* gene which controls the *Il4/Il13/Il5* Th2 cytokine locus, and also *Il17a*, *Il12rb2*, and *Tbx21*. The gray boxes indicate IL-2 pDHSs which bind STAT5 in Th0 cells and other STAT proteins in other T cell lineages.

therefore not be true pioneer factors for the activation of whole loci, but initially home in on sites rendered accessible by IL-2 signaling and chromatin reprogramming at an earlier stage during blast cell transformation. A similar notion was recently proposed in a study of the human *RORC* locus whereby TCR signaling to NFAT initially promotes chromatin accessibility in this locus prior to terminal differentiation, thereby creating a permissive environment for factors induced by polarizing cytokines (Yahia-Cherbal *et al*, 2019).

In support of this model, an increasingly important role for IL-2 during CD4 and CD8 memory T cell formation has now emerged, both for the initial priming of genes during the first phase of T cell activation (Dooms *et al*, 2004, 2007; Williams *et al*, 2006; Pepper *et al*, 2011), and for maintaining immunity when effector T cells revert to memory T cells (McKinstry *et al*, 2014). Indeed, memory T cell recall was impaired in *Listeria monocytogenes*-infected mice in CD4 T cells not expressing the IL-2 receptor on day 3 of the primary response (Pepper *et al*, 2011). An independent study using LCMV-infected mice showed that IL-2 was required during the first 6 days of infection in order for CD8 memory T cells to mount an efficient response to the secondary infection (Williams *et al*, 2006). More recently, IL-2 has also been shown to be required for the formation of tissue-resident memory T cells (T$_{RM}$) in the lung (Hondowicz *et al*, 2016). The need for IL-2 exposure early during T cell activation suggests an IL-2-dependent priming mechanism causing heritable changes to the epigenome which are maintained through T cell expansion, differentiation, contraction, and homeostasis. This idea is consistent with several recent studies which suggest that memory T cells are derived from effector T cells which have been previously activated and have subsequently proliferated (Akondy *et al*, 2017; Youngblood *et al*, 2017) rather than formed from a population of naïve T cells which have bypassed antigenic activation (Restifo & Gattinoni, 2013). However, IL-2 alone cannot by itself account for the long-term maintenance of AP-1/STAT-dependent immunological memory. IL-2 is a T cell-derived growth factor that requires ongoing TCR signaling for its expression. Hence, the role of IL-2 in locus priming would need to be replaced by other factors such as the γc cytokine IL-7, which is expressed by stromal cells in lymph nodes, the γc cytokine IL-4 expressed by Th2 cells (Ben-Sasson *et al*, 1990), and IL-15 which is important for tissue-resident memory T cells (Mackay *et al*, 2015). Consistent with this model, we demonstrate here that IL-7 (and by inference other γc cytokines such as IL-15 and IL-4 as well) can functionally substitute for IL-2 to maintain epigenetic priming of pDHSs occupied by AP-1 and STAT5 in recently activated T$_B$ cells.

**Chromatin priming enables specific gene expression programs**

We previously defined pDHSs as a discrete class of gene regulatory element, distinct from classically defined transcriptional enhancers, which function in T cells to maintain active chromatin domains in an accessible chromatin conformation while having little impact on steady-state levels of transcription (Bevington *et al*, 2016, 2017a). Chromatin priming is a key mechanism contributing to the regulation of developmental and lineage-specific gene expression in many cell types and contexts, ensuring that genes are able to respond to signals and that they are expressed at the right time and place (Bonifer & Cockerill, 2017). During development, genes are typically activated in a stepwise fashion, and chromatin priming often occurs prior to the actual onset of transcription (Gualdi *et al*, 1996; Kontaraki *et al*, 2000; Walter *et al*, 2008; Mercer *et al*, 2011; Wang *et al*, 2015; Goode *et al*, 2016; Huang *et al*, 2016). For example, during macrophage development, PU.1 binding provides the cue and creates the accessibility for additional factors to bind to macrophage-specific or inducible genes (Ghisletti *et al*, 2010; Heinz *et al*, 2010; Glass & Natoli, 2016). Also in macrophages, many enhancers remain in an inactive state until they are activated for the first time by STAT and AP-1 proteins and then persist in a primed latent state hypersensitive to reactivation (Ostuni *et al*, 2013).

We previously reported that NFAT is a major mediator of cyclosporin A-sensitive chromatin opening downstream of TCR signaling at enhancers which require cooperation between both NFAT and AP-1 (Cockerill *et al*, 1993; Johnson *et al*, 2004). More recently, a dominant-negative derivative of FOS has been used to show that AP-1 is also essential for chromatin opening downstream of TCR signaling (Yukawa *et al*, 2019). This activity of AP-1 may now help to explain how IL-2 signaling is also able to maintain chromatin accessibility at primed genes once they have been rendered accessible by prior TCR signaling.

As a class of DNA elements in T cells, pDHSs were previously thought to be formed by a hit-and-run mechanism whereby transient expression of AP-1 induced by TCR signaling was required to cooperate with other factors to initially open up nucleosome-free regions that could then be stably maintained by constitutively expressed factors such as RUNX1 and ETS1. Here, we now show that IL-2-dependent pDHSs represent a subset of this class of regulatory element which are created by TCR and IL-2 signaling, but require subsequent reinforcement from γc cytokine-inducible AP-1 and STAT5 for their maintenance. In this model, IL-2 and/or IL-7 function as true homeostatic factors whereby they are insufficient to induce immune response genes, and may even suppress some programs until the right factor is encountered. Evidence suggests that the IL-2-inducible AP-1 family member JUND suppresses rather than activates immune responses (Meixner *et al*, 2004), in keeping with the low levels of gene expression seen at many primed genes in the absence of TCR-inducible FOS and JUN expression (Bevington *et al*, 2016, 2017a). Furthermore, the forced expression of JUND suppresses activation of the Th2 gene expression program (Meixner *et al*, 2004). Differentiation along the Th17 pathway may also

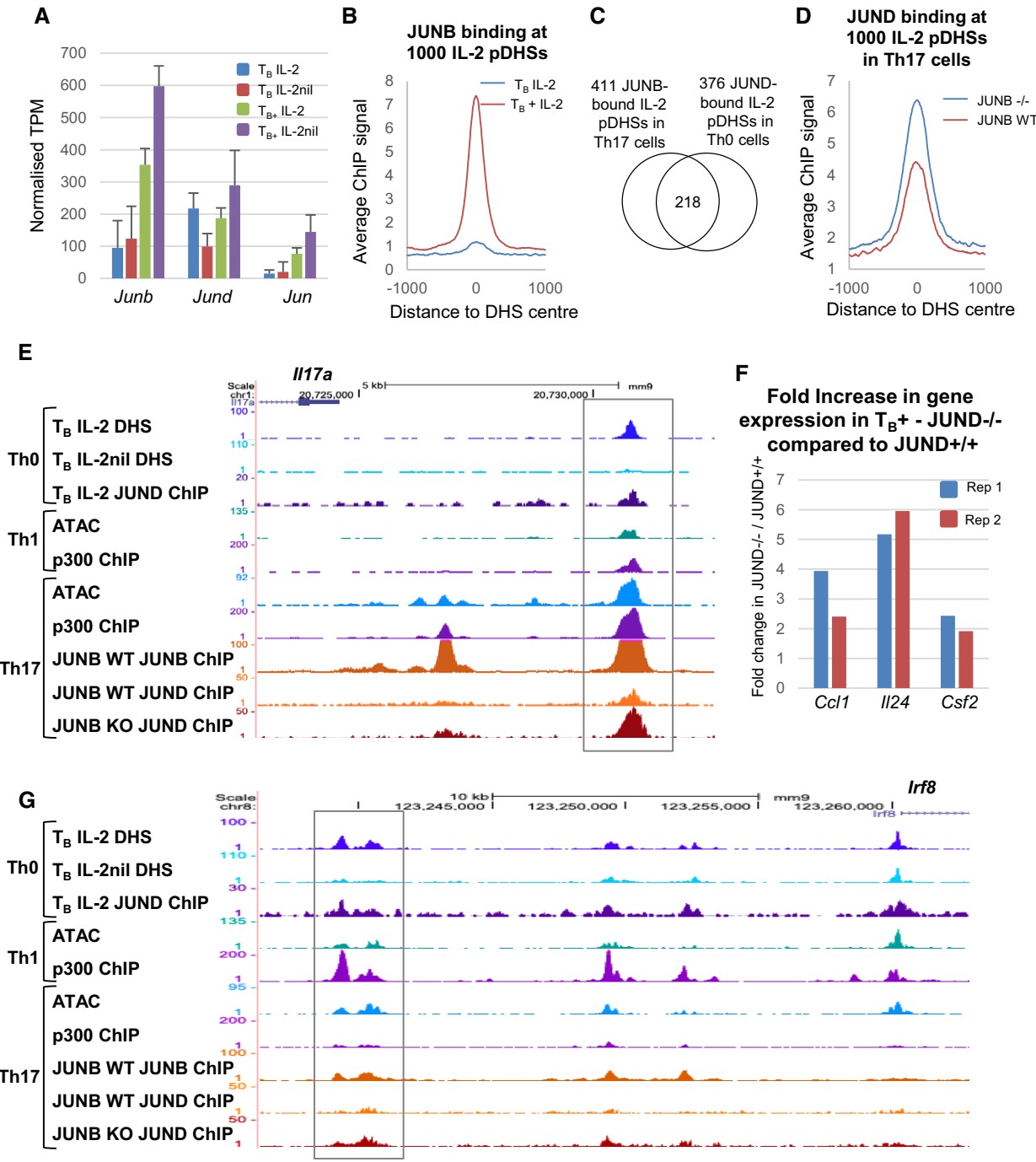

**Figure 10. JUNB and JUND bind to IL-2 pDHSs in multiple T cell lineages.**

A mRNA expression levels of JUN family transcription factors. Normalized TPM is shown for T_B IL-2, T_B IL-2nil, T_B+ IL-2, and T_B+ IL-2nil. Standard deviation is shown for 3 replicates.

B Average JUNB ChIP-Seq signal at the 1,000 IL-2 pDHSs in T_B IL-2 and T_B+ IL-2 (Bevington et al, 2016).

C Overlap of the 411 JUNB bound IL-2 pDHSs in Th17 cells (Carr et al, 2017) with the 376 JUND bound IL-2 pDHSs in T-blast (Th0) cells.

D Average JUND ChIP-Seq signal at the 1,000 IL-2 pDHSs in Junb^{+/+}CD4^{Cre} (Junb WT) and Junb^{f/f}CD4^{Cre} (Junb KO) Th17 cells.

E Genome browser tracks showing open chromatin (DNase-Seq or ATAC-Seq) and ChIP-Seq in Th0, Th1, and Th17 cells at the Il17a locus (Ciofani et al, 2012; Vahedi et al, 2012; Carr et al, 2017). The gray box indicates a DHS which binds JUND.

F Fold change in gene expression in T_B+ IL-2 cells for Jund^{−/−} compared to Jund^{+/+} mice. Data from 2 replicates are shown.

G Genome browser tracks showing open chromatin (DNase-Seq or ATAC-Seq) and ChIP-Seq in Th0, Th1, and Th17 cells at the Irf8 locus. The gray box indicates an IL-2 pDHS which binds JUND.

initially be enabled by IL-2-dependent priming of Th17-specific genes, such as IL-17α, but then stalled because expression of IL6Rα is suppressed by IL-2, and IL-17α expression is dependent on the IL-6-dependent induction of STAT3 (Liao *et al*, 2011a,b; Yang *et al*, 2011). Therefore, Th17 cells may require a relative loss of IL-2 signals to fully differentiate.

Taken together, our data suggest that the basic framework underlying lineage-specific effector T cell gene expression programs is laid down during the earliest stages on T cell activation and is kickstarted by TCR/IL-2-dependent signaling processes. This set of signals is sufficient to enable Th0 cells to progress down distinct alternative developmental pathways, depending which environment they encounter. Once established, the priming of immune response genes can be maintained in the long term by homeostatic cytokines in effector T cells once these cells return to quiescence as memory T cells. This type of progressive activation of gene loci, via cooperation between functionally distinct classes of regulatory element, is likely to be a common theme throughout development (Bonifer & Cockerill, 2017).

# Materials and Methods

## Experimental

### Mice

C42 transgenic mice were described previously (Mirabella *et al*, 2010). C42 mice contain a 130-kb DNA fragment of the human *IL3/CSF2* locus. The mice were back-crossed with C57 Black 6 mice (C57BL/6J) for over 10 generations. To track Th17 cells *in vivo*, *Il17a*$^{cre}$ mice (Hirota *et al*, 2011), crossed to *ROSA26*$^{tdRFP}$ mice (Luche *et al*, 2007), were used to fate-map IL-17a expression. To inducibly delete *Il7r in vivo*, *CD4*$^{creERT2}$ mice (Aghajani *et al*, 2012) (obtained from JAX) were crossed with *Il7r*$^{f/f}$ mice (McCaughtry *et al*, 2012) and then additionally crossed with *ROSA26*$^{tdRFP}$ mice (Luche *et al*, 2007) to aid detection of cre activity. *Jund*$^{-/-}$ mice were described previously (Thepot *et al*, 2000) and maintained as *Jund*$^{+/-}$ mice for breeding. Animals were used in accordance with Home Office guidelines at the University of Birmingham. Mice were housed at 21°C ± 2°C, 55% humidity (± 10%) with 12-h light dark/cycle in 7-7 IVC caging with environmental enrichment of plastic houses plus paper bedding.

### T-blast cell culture, purification, and stimulation

Actively dividing T-blast cells (T$_B$) were generated from freshly isolated CD4 T cells purified from the spleens of C42, *Jund*$^{+/+}$, or *Jund*$^{-/-}$ mice using MACS CD4 (L3T4) microbeads (Miltenyi). Cells were cultured at $1 \times 10^6$ cells/ml in IMDM with 2 μg/ml Concanavalin A for 40 h before the Concanavalin A was removed, and the cells were maintained at $5 \times 10^5$ cells/ml in IMDM supplemented with 50 U/ml recombinant murine IL-2 or IL-7 (PeproTech). To allow for sufficient expansion, the cells were cultured for 30 h in the presence of cytokine and then were either maintained with IL-2 or IL-7 alone, maintained in IL-2 with the addition of 1 μM Ruxolitinib (STEMCELL Technologies) or were washed twice and resuspended in IMDM alone. Cells were cultured for a further 16 h, and where required, dead cells were removed using the MACs dead cell removal kit (Miltenyi #130-090-101) and analyzed for apoptotic cells

using FITC Annexin V Apoptosis Detection Kit I (BD Pharmingen #556547). Cells were either harvested immediately or stimulated for 2 h with 20 ng/ml phorbol myristate acetate and 2 μM calcium ionophore A23187 (PMA/I) at $1 \times 10^6$ cells/ml in IMDM before processing for experiments. For control purposes in Fig 9A, a parallel method of generating T cell blasts was employed whereby freshly isolated spleen CD4 T cells were cultured at $1 \times 10^6$ cells/ml in IMDM with CD3 and CD28 antibodies for 40 h before maintenance at $5 \times 10^5$ cells/ml in IMDM supplemented with 50 U/ml recombinant murine IL-2 for 48 h.

### Antigen-specific Th1 memory T cell generation and purification

C42 mice were infected with $10^7$ *actA*-deficient *L. monocytogenes* expressing OVA-2W1S (*Lm*-2W1S, kind gift from Dr. M. Jenkins) through intravenous injection in the tail vein. Glycerol stocks of *Lm*-2W1S, aliquoted into 1 ml cryovials, were centrifuged and washed with sterile PBS before resuspending in adequate volume of sterile PBS to obtain $10^7$ bacteria per 200 μl.

To recover 2W1S-specific CD4 T cells, spleens from infected mice were recovered 56 dpi, crushed through a nylon mesh, and treated with Gey's solution red blood lysis buffer. The 2W1S-specific memory T cell pool was enriched by magnetic enrichment, as described by Jenkins (Moon *et al*, 2009). The resulting cells were stained with CD3-PE, CD4-PE-Cy7, CD44-FITC, B220-e450, CD11c-e450, CD11b-e450, and 2W1S:A$^b$-APC. CD44$^{hi}$ 2W1S-specific CD4 T cells were sort purified using a BD FACSAria Fusion cell sorter (BD) to greater than 95% purity.

### Antigen-specific memory T cell generation and purification of *Il7r*$^{+/+}$ and *Il7r*$^{-/-}$ Th1 cells

*CD4*$^{creERT2}$ × *Il7r*$^{f/f}$ × *ROSA26*$^{tdRFP}$ or *Il7r*$^{f/f}$ × *ROSA26*$^{tdRFP}$ (littermate control) mice were infected as described above. 19–21 days later, 200 μl 20 mg/ml tamoxifen was administered by oral gavage for four consecutive days. 2W1S-specific CD4 T cells were recovered from infected mice 14–18 days following the first treatment with tamoxifen and processed as above. The resulting cells were stained with CD3-AF700, CD4-PerCP-Cy5.5, CD8-BV510, B220-APC-Cy7, CD11c-APC-Cy7, CD11b-APC-Cy7, CD44-BV786, IL-7Rα-BV421, and 2W1S:A$^b$-APC. The *Il7r*$^{f/f}$ × *ROSA26*$^{tdRFP}$ samples were sort purified as CD44$^{hi}$ IL7Rα+ 2W1S-specific CD4 T cells, and the *CD4*$^{creERT2}$ × *Il7r*$^{f/f}$ × *ROSA26*$^{tdRFP}$ samples were purified as CD44$^{hi}$ tdRFP$^+$ IL-7Rα- 2W1S-specific CD4 T cells. Cells were sorted using a BD FACSAria Fusion cell sorter to greater than 95% purity.

### Generation of Th17 cells in vivo

To generate and label Th17 cells *in vivo*, *Il17a*$^{Cre}$ × *ROSA26*$^{tdRFP+}$ mice (kind gift from Dr. G Frankel) were orally infected with ~ $2 \times 10^9$ *Citrobacter rodentium* strain ICC169 in 200 μl PBS by oral gavage 7 days before harvesting Th17 cells. Following fat removal, colons were cut longitudinally into 0.5-cm sections and washed with Hank's Balanced Salt Solution (HBSS, Gibco) with 2% fetal bovine serum. Tissue was incubated twice with HBSS 2 mM EDTA at 37°C for 15 min and then digested in complete RPMI with 0.85 mg/ml Collagenase V (Sigma), 1.25 mg/ml Collagenase D (Roche), 1 mg/ml Dispase (Gibco), and 30 μg/ml DNAse I (Roche) at 37°C until the connective tissue was fully digested (approximately 45 min). Cells were passed through 100- and 70-μm strainers and stained with LIVE/DEAD™ Fixable Near-IR stain (Invitrogen) for 20 min on

ice prior to antibody staining. Antibody staining was carried out on ice for 1 h. Colon-derived cells were stained with antibodies raised against the mouse antigens: CD45.2 (clone 104), B220 (clone RA3-6B2), CD11b (clone M1/70), CD11c (clone N418), CD3 (clone 17A2), CD4 (clone RM4-5), CD8a (clone 58-6.7), and CD44 (clone IM7). Live Th17 cells were purified as tdRFP$^+$ CD4 T cells (fate mapping IL-17α expression) using a BD FACSAria Fusion cell sorter (BD) to greater than 95% purity.

### DNase I hypersensitive site analysis

DNase I digestions were performed as previously described (Bert *et al*, 2007). Briefly, permeabilized cells were digested at $5 \times 10^6$ cells/ml for 3′ at 22°C in DNase I buffer (60 mM KCl, 15 mM NaCl, 5 mM MgCl$_2$, 100 mM Tris–HCl pH 7.4, 1 mM EGTA pH 7.4, 0.3 M sucrose, 0.2% NP40, 1 mM CaCl$_2$, and DNase I). Reactions were terminated by SDS (final concentration 0.5%) and treated with 0.5 mg/ml Proteinase K at 37°C overnight followed by 0.2 mg/ml RNase A for 1 h. DNA was purified by phenol/chloroform extraction. A range of DNase I concentrations were used, and the optimal digestion for DNase-Seq library preparation was determined as described previously (Bevington *et al*, 2016).

### Western blot analysis of proteins

Nuclear extracts were prepared using the Nuclear Complex Co-IP Kit according to the manufacturer's instructions (Active Motif #54001). Protein extracts were separated by electrophoresis on 4-20% Mini-PROTEAN® TGX™ gels (Bio-Rad #4561096) in TGS (25 mM Tris, 190 mM glycine, 0.1% SDS) before transfer to nitrocellulose membranes using a Trans-Blot® Turbo™ Transfer system (Bio-Rad). Membranes were blocked with 5% (*w*/*v*) milk powder in TBS-Tween (TBST) (20 mM Tris pH 7.5, 150 mM NaCl, 0.1% Tween) at room temperature for 1 h before incubation overnight at 4°C with 1:1,000 dilution of primary antibodies in 5% (*w*/*v*) milk powder in TBST. Primary antibodies: JUND (Thermo Scientific #720035), p-STAT5 (Y694) (Cell signaling #9351S) STAT5 (Santa Cruz #sc-835X), RUNX1 (Abcam #ab23980), B2M (Abcam #ab75853), and ETS1 (Santa Cruz #sc-350X). Membranes were washed with TBST and incubated with 1:10,000 dilution of HRP linked anti-rabbit IgG secondary antibody (Santa Cruz #sc-2054) for 1 h at room temperature. Membranes were visualized using the SuperSignal® West Pico Chemiluminescent Substrate (Thermo Scientific #34577).

### Chromatin immunoprecipitation

Chromatin Immunoprecipitation (IP) was carried out as described previously using a double-cross-linking method (Bevington *et al*, 2016). Cells were resuspended at $3.3 \times 10^6$ cells/ml and cross-linked for 45 min with 0.83 mg/ml Di(N-succinimidyl) glutarate (DSG—Sigma). Cells were washed four times with PBS before resuspending at $2 \times 10^6$ cells/ml and re-cross-linking for 10 min with 1% formaldehyde (Pierce) in PBS. Reactions were quenched with 0.125 M glycine, washed twice with PBS, and incubated with Buffer A (10 mM HEPES pH 8.0, 10 mM EDTA, 0.5 mM EGTA, 0.25% Triton X-100) at 4°C for 10 min followed by Buffer B (10 mM HEPES pH 8.0, 200 mM NaCl, 1 mM EDTA, 0.5 mM EGTA, 0.01% Triton X-100, 0.25% SDS) at 4°C for 10 min. Chromatin was sheared ($10^7$ cells/ml) in IP Buffer I (25 mM Tris 1 M, pH 8.0, 150 mM NaCl, 2 mM EDTA, pH 8.0, 1% Triton X-100, and 0.25% SDS) using a Picorupter™ (Diagenode) to fragments of ∼ 500 bp. The supernatant

was collected and diluted 3-fold in IP Buffer II (25 mM Tris, pH 8.0, 150 mM NaCl, 2 mM EDTA, pH 8.0, 1% Triton X-100, 7.5% glycerol). Chromatin from $2 \times 10^6$ cells was incubated with 2 μg JUND (Santa Cruz #sc-74X) or STAT5 (Santa Cruz #sc-835X) coupled to 15 μl protein G Dynabeads (Dynal) for 2 h at 4°C. Beads were washed with Buffer 1 (20 mM Tris–HCl, pH 8.0, 150 mM NaCl, 2 mM EDTA, pH 8.0, 1% Triton X-100, 0.1% SDS), twice with Buffer 2 (20 mM Tris–HCl pH 8.0, 500 mM NaCl, 2 mM EDTA, pH 8.0, 1% Triton X100, 0.1% SDS), LiCl buffer (10 mM Tris–HCl, pH 8.0, 250 mM LiCl, 1 mM EDTA, pH 8.0, 0.5% NP-40, 0.5% Na-deoxycholate) and twice with TE/NaCl buffer (10 mM Tris–HCl pH 8.0, 50 mM NaCl, 1 mM EDTA). Samples were eluted in 100 μl elution buffer (100 mM NaHCO$_3$, 1% SDS), and crosslinks were reversed overnight at 65°C with proteinase K. DNA was purified using Agencourt AMPure (Beckman Coulter) according to the manufacturer's instructions. Samples were analyzed and validated by qPCR.

### Gene expression analyses

Total RNA was extracted from $1 \times 10^6$ cells using TRIzol™ (Invitrogen) according to the manufacturer's instructions. Contaminating DNA was removed using Turbo DNA-free™ kit (Ambion). For RNA-Seq, samples were further purified using the RNeasy® MinElute Cleanup Kit (Qiagen). First-strand cDNA synthesis was carried out using Superscript II Reverse Transcriptase and Oligo dT (Invitrogen) according to the manufacturer's instructions. Expression levels were measured by qPCR using Applied Biosystems SYBR Green master mix. Statistics were generated from at least three samples using a paired *t*-test. The PCR primers used were as follows:

**B2m**: 5′-TTCTGGTGCTTGTCTCACTG 5′-CAGTATGTTCGGCTTCCC ATTC

**Il2**: 5′-GATGAACTTGGACCTCTGCG 5′-CATCATCGAATTGGCACT CA

**Csf2**: 5′-ATCAAAGAAGCCCTGAACCTCC 5′-CCCGTAGACCCTGCTC GAATAT

**Il24**: 5′-GGAACAGCAAAAACCGCTGG 5′-GCCAACAACTTCATAGT-CATCATGTC

**Ifng**: 5′-TCAAGTGGCATAGATGTGGAC 5′-TGAGGTAGAAAGAGA-TAATCTGG

**Il13**: 5′-GCAGCATGGTATGGAGTGTG 5′-TGGCGAAACAGTTGCTT TGT

**Ccl1:** 5′-CCAGACATTCGGCGGTTG 5′-CAGCAGCAGGCACATCAG

### Assay for transposase accessible chromatin using sequencing (ATAC-Seq)

The omni ATAC-Seq protocol was modified from Corces *et al* (2017). Briefly, 2W1S-specific T cells were purified by flow cytometry from *CD4*$^{creERT2}$ × *Il7r*$^{f/f}$ × *ROSA26*$^{tdRFP}$ mice (3,000–11,000 IL-7Rα-deficient cells) or from *Il7r*$^{f/f}$ × *ROSA26*$^{tdRFP}$ mice (8,000–20,000 IL-7Rα$^+$ cells). IL-17a fate-mapped Th17 cells were purified by flow cytometry from *Il17a*$^{Cre}$ × *ROSA26*$^{tdRFP+}$ mice (50,000 tdRFP$^+$ cells). Cells were resuspended in 50 μl ATAC transposition reaction mix containing 25 μl 2× Tagment DNA Buffer (Illumina), 2.5 μl Tn5 transposase (Illumina), 16.5 μl PBS, 0.5 μl 10% Tween-20, and 0.5 μl 1% Digitonin (Promega #G9441) and incubated for 20 min at 37°C. For the 2W1S-specific Th1 memory cells, 40,000 cells were resuspended in 50 μl ATAC transposition reaction mix containing 25 μl 2× Tagment DNA Buffer (Illumina), 2.5 μl Tn5 transposase (Illumina), and 0.5 μl 1% Digitonin and incubated for

30 min at 37°C. DNA was purified using the MinElute Reaction Clean up kit (Qiagen #28204) before PCR amplification using Nextera custom primers. Optimal amplification was achieved by a qPCR side reaction as described (Corces *et al*, 2017). Amplified DNA was subsequently purified using Ampure Beads (Beckman Coulter) prior to validation. Libraries were sequenced on NextSeq® 500/550 High Output kit v2 75 cycles (Illumina, FC 404-2005) at the Genomics Birmingham sequencing facility.

### Library preparation

DNA libraries from DNase I digestions and chromatin immunoprecipitation assays were prepared using the Kapa Hyper-Prep kit according to the manufacturer's guidelines. Libraries were indexed and then amplified by PCR. For DNase-Seq, libraries were amplified for 12 PCR cycles, and for ChIP-Seq, libraries were amplified for 16 cycles. DNA fragments from amplified libraries were agarose gel electrophoresis purified to ensure the correct fragment lengths were sequenced (Qiagen Gel purification Kit). Fragments of 200–300 bp were selected for DNase-Seq libraries and 200–400 bp for ChIP libraries. Libraries were validated by qPCR and quantified using the Kapa quantification kit before pooling and sequencing on NextSeq® 500/550 High Output kit v2 75 cycles (Illumina, FC 404-2005).

RNA libraries were prepared from 50 to 100 ng RNA using the NEB Next® Ultra™ II Directional RNA library Prep kit for Illumina (NEB, #E7760S) according to the manufacturer's instructions. rRNA was depleted using the NEBNext® rRNA Depletion kit (NEB, #E6310S). Libraries were quantified using the Kapa quantification kit before pooling and sequencing on NextSeq® 500/550 High Output kit v2 150 cycles (Illumina, FC 404-2002).

### Data analysis

#### Alignment, coverage, and peak detection of DNase-Seq, ATAC-Seq, and ChIP-Seq data sets

Raw DNA sequencing reads were aligned to the NCBI and the Mouse Genome Sequencing Consortium version mm9 Build 37 from July 2007 using *bowtie2* (Galaxy version 2.3.2.2) (Langmead & Salzberg, 2012) with the preset –very-sensitive-local. Duplicated reads were removed from ChIP-Seq data using the Picard tool MarkDuplicates (Galaxy Version 2.7.1.1). Coverage files were generated using MACs version 1.4.2 using –g mm –keep-dup auto –w -S as parameters (Zhang *et al*, 2008).

#### Normalization of DNase-Seq data sets

To ensure that the most accurate coordinates were calculated for each peak summit and to determine a complete set of summits for all samples, the BAM files from the DNase-Seq experiments $T_B$ IL-2, $T_B$ IL-2nil, $T_B+$ IL-2, $T_B+$ IL-2nil $T_B$ IL-7, and $T_B+$ IL-7 were merged using BamTools (Galaxy version 0.0.2). MACS2 callpeak (Galaxy version 2.1.1) was then used to identify 91,313 summits across all samples. The DNA sequence tags ± 200 bp from the peak summit were counted for each individual sample using the annotatePeaks function of the HOMER package (Heinz *et al*, 2010). The samples were normalized to one another using a correction factor based on the median of the top 25,000 peaks. Correction factors were then used to normalize genome browser scales, average profiles, and contrast levels in tag density profiles.

### Normalization of ATAC-Seq data sets

A high confidence set of peaks was determined for the for the 3 $CD4^{\text{creERT2}} \times Il7r^{\text{f/f}} \times ROSA26^{\text{tdRFP}}$ samples and the 3 $Il7r^{\text{f/f}} \times ROSA26^{\text{tdRFP}}$ samples by intersecting the peaks determined by MACs (summit files ± 200 bp) using the BEDTools v2.26.0 intersect function (Quinlan & Hall, 2010). The samples were normalized to one another using a correction factor based on the median of the high confidence peaks. Reads were counted ± 200 bp from the peak summit using the annotate peak function of the HOMER package (Heinz *et al*, 2010). Correction factors were used to normalize genome browser scales and average profiles.

### Normalization of ChIP data sets

ChIP samples were normalized according to the number of mapped reads. These values were then used to normalize average profiles and contrast levels in heatmaps.

### Unions of DNase-Seq samples

DNase-Seq DNA sequence tags were counted in each sample at the 91,313 merged summits ± 200 bp. The significant peaks were then identified using a cut-off to exclude background and insignificant peaks. Two samples were compared by merging the peaks using the sort and merge function of the bedtools package. This gave a union of peaks with common and specific sites for each sample. The normalized number of DNA sequence tags ± 200 bp from the summits of the union was then compared between the two samples to give a fold change difference. The data were visualized as sequence tag density plots showing the aligned reads for two different samples at both common and unique regions ordered according to the fold change difference in tag density of one sample compared to the other.

### DNA sequence tag density profiles

DNA sequence tag density profiles were generated using the annotatePeaks function of the HOMER package using -hist 10 -ghist -size 2000 as parameters. Images were generated using Java Treeview (http://jtreeview.sourceforge.net/).

### Average DNA sequence tag density plots

Average DNase-Seq, ATAC-Seq, and ChIP-Seq DNA sequence tag density profiles were generated ± 1 kb around the DHS summit using the annotatePeaks function of the HOMER package with -hist 10 -size 2000 as parameters.

### Intersection of peak groups

Where duplicate samples were available, high confidence peak sets were defined by overlapping the peak summits determined by MACs (± 200 bp) using BEDTools v2.26.0 intersect function (Quinlan & Hall, 2010) before intersecting with other peak groups.

### Motif discovery

*De novo* DNA motif analysis was performed using the findMotifsGenome.pl function of the HOMER package (Heinz *et al*, 2010). Motifs were identified ± 100 bp from the peak summit. STAT5-binding motifs were located within the IL-2 pDHSs using the AnnotatePeaks.pl function of the HOMER package.

### Distance analyses

The distance between the IL-2 pDHSs and IL-2 iDHSs was calculated using the BEDTools closestBed function (Galaxy Version 2.26.0.0).

### Public data sets

DNase-Seq data sets CD4 $T_M$, CD4 $T_N$, CD4 $T_N+$ (Bevington et al, 2016), GEO accession number GSE67451. Th2 ATAC-Seq (Shih et al, 2016), GEO accession number GSE77695, Tfh ATAC-Seq (Chen et al, 2020) GEO accession number GSE120532, Treg ATAC-Seq (Garg et al, 2019), GEO accession number GSE121764. CD4 $T_B$ RUNX1 and ETS1 ChIP-Seq data sets (Bevington et al, 2016), GEO accession number GSE67443. Th2 GATA3 ChIP-Seq (Wei et al, 2011), GEO accession number GSE20898. Th2 STAT6 ChIP-Seq (Wei et al, 2010), GEO accession number GSE22105. Th1 T-BET ChIP-Seq (Gokmen et al, 2013), GEO accession number GSE40623. Th1 STAT4 ChIP-Seq (Wei et al, 2010), GEO accession number GSE22105. Th17 STAT3 ChIP-Seq (Durant et al, 2010), GEO accession number GSE21669. Th17 RORγt ChIP-Seq and Th17 p300 ChIP-Seq (Ciofani et al, 2012), GEO accession number GSE40918. JUNB ChIP-Seq and JUND ChIP-Seq in $Junb^{+/+}CD4^{Cre}$ and $Junb^{fl/fl}CD4^{Cre}$ Th17 cells (Carr et al, 2017), GEO accession number GSE98414. WT Th1 p300 ChIP-Seq (Vahedi et al, 2012), GEO accession number GSE40463. Data were downloaded and processed as described above.

### RNA-Seq data analyses

Raw paired-end cDNA library sequence reads were processed prior to alignment with Trimmomatic v0.32 (Bolger et al, 2014) in order to remove any low-quality sequences and adapters. The processed DNA sequence reads were then aligned to the mouse genome (version mm9) using Hisat2 v2.1.0 (Kim et al, 2015) with default parameters. Gene expression levels were measured as Transcripts Per Million (TPM) calculated with StringTie v1.3.3 (Pertea et al, 2015) using RefSeq gene models as the reference transcriptome. Only genes that could be detected with a TPM value greater than 1 in at least one of the samples were retained for further analysis. TPM values were normalized using upper-quartile normalization in R v3.5.1 and further $\log_2$-transformed as $\log_2(TPM + 1)$. Differential gene expression analysis was carried using the limma package v3.38.3 (Ritchie et al, 2015) in R. A gene was considered to be significantly differentially expressed if it had a greater than 3-fold change between experimental conditions, and a Benjamini-Hochberg adjusted $P$-value $< 0.05$.

Principal components analysis (PCA) was carried out with the $\log_2$-transformed gene expression values using the prcomp() function in R. Hierarchical clustering was carried out by first calculating the Pearson correlation of the $\log_2$-transformed TPM values for each pair of samples. These values were then clustered using complete-linkage clustering of the Euclidean distances in R.

## Data availability

All DNA sequence data generated for this study are deposited in the GEO depository under GEO accession GSE147294 (http://www.ncbi.nlm.nih.gov/geo/query/acc.cgi?acc=GSE147294).

**Expanded View** for this article is available online.

## Acknowledgements

This study was supported by funding from the Medical Research Council (MR/P001319/1). We thank Constanze Bonifer her input in the preparation of the manuscript. We thank Genomics Birmingham at the University of Birmingham for assistance with DNA sequencing. We thank Csilla Varnai for assistance preparing the GEO submission of the genome-wide sequencing data. We also thank the following for their kind provision of mouse strains: Ute Bank ($Il7r^{f/f}$ mice), Jorg Fehling ($ROSA26^{tdRFP}$ mice), Gitta Stockinger ($Il17a^{Cre}$ mice), and Brigitte Bourachot and Fatima Mechta-Grigoriou ($JunD \pm$ mice). The following tetramer was obtained through the NIH Tetramer Facility: 2W1S:I-A[b].

## Author contributions

SLB, JKS, ST, DWG, RF, VM-R, and CMW performed the experiments. SLB, DRW, and PNC wrote the manuscript. SLB and PK analyzed the data.

## Conflict of interest

The authors declare that they have no conflict of interest.

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
