## [Review Process File · The EMBO Journal]

IL-2/IL-7-inducible factors pioneer the path to T cell differentiation in advance of lineage-defining factors

Sarah Bevington, Peter Keane, Jake Soley, Saskia Tauch, Dominika Gajdasik, Remi Fiancette, Veronika Matei-Rascu, Claire Willis, David Withers, and Peter Cockerill

DOI: [10.15252/embj.2020105220](https://doi.org/10.15252/embj.2020105220)

Corresponding authors: Peter Cockerill (p.n.cockerill@bham.ac.uk), Sarah Bevington (s.l.bevington@bham.ac.uk), David Withers (d.withers@bham.ac.uk)

Review Timeline:

Submission Date:	8th Apr 20
Editorial Decision:	14th May 20
Revision Received:	6th Jul 20
Editorial Decision:	5th Aug 20
Revision Received:	13th Aug 20
Accepted:	17th Aug 20

Editor: Karin Dumstrei

Transaction Report:

Dear Peter,

Thank you for submitting your manuscript to the EMBO Journal. Your study has now been seen by three referees and their comments are provided below.

As you can see from the comments the referees appreciate that the analysis adds insight into a role of IL-2 in T cell fate differentiations. Referee #1 feels that the analysis would be strengthened if analysis using in vivo models could be included but also see that there is a value to the current dataset. Given the comments provided I would like to invite you to submit a revised manuscript that takes the raised concerns into consideration. We don't need further in vivo data. I think it would be helpful to discuss the revisions further and we can do so either via video/phone or email - whatever you prefer.

Thank you for the opportunity to consider your work for publication. Looking forward to discussing the revisions further.

with best wishes

Karin

Karin Dumstrei, PhD
Senior Editor
The EMBO Journal

When assembling figures, please refer to our figure preparation guideline in order to ensure proper formatting and readability in print as well as on screen:
<http://bit.ly/EMBOPressFigurePreparationGuideline>

- a point-by-point response to the referees' comments, with a detailed description of the changes made (as a word file).
- a word file of the manuscript text.

- individual production quality figure files (one file per figure)
 - a complete author checklist, which you can download from our author guidelines (<https://www.embopress.org/page/journal/14602075/authorguide>).
 - Expanded View files (replacing Supplementary Information)
- Please see out instructions to authors
<https://www.embopress.org/page/journal/14602075/authorguide#expandedview>

The revision must be submitted online within 90 days; please click on the link below to submit the revision online before 12th Aug 2020.

Link Not Available

Referee #1:

In this study, Bevington et al measure chromatin accessibility of in vitro activated CD4 T cells and show that withdrawal of IL-2 results in loss of chromatin accessibility as measured by DNase hypersensitivity (DHS) with preservation by inclusion of IL-7. Blocking cytokine signaling with the JAK inhibitor, ruxolitinib, also reduces accessibility, but not as effectively as withdrawal of IL-2. Motif analysis of accessible regions revealed enrichment of ETS, STAT, and AP-1 motifs. The authors further show that JunD expression is dependent upon IL-2. Chip-seq analysis shows that STAT5 and JunD broadly bind to all regions of accessibility and are dependent upon IL-2. The authors next stimulate T blasts cultured with or without IL-2 with PMA/ionomycin and identify loss of 2578 of 38645 peaks when IL-2 is withdrawn; again, IL-7 preserved accessibility. As expected, NFAT and NFAT/AP1 composite peaks were enriched with PMA/Ionomycin stimulation. The authors go on to show that gene expression and accessibility correlate; although, IL-2 withdrawal only affected 14 genes. Given this modest effect, the authors compared their IL-2-dependent accessible sites to public Chip-seq datasets, noting that the 62.5% of IL-2-regulated accessible peaks overlapped with other STATs and "master regulator" TF binding in different helper cell subsets. The authors conclude that IL-2 signaling is required to maintain chromatin accessibility at hundreds of regulatory elements, including key subset-specific genomic regions, supporting T cell memory.

The authors conclusions are provocative and interesting; however, the study relies heavily upon in vitro stimulation using artificial conditions (concanavalin A, PMA/ionomycin etc). The authors do not test their hypotheses using solid in vivo systems, and in this respect, the scope of the manuscript is necessarily limited. The mechanistic role of IL-2 in memory cells is an important topic; but the conclusions from the present study are largely inferential. The idea that IL-2/STAT5 provide critical "pioneer" functions in chromatin accessibility and priming of gene expression is exciting; however,

the impact of the findings would be greater if robust in vivo models were included.

Activated T cells produce IL-2; this raises the issue as to whether the results might be more dramatic if anti-IL-2/anti-IL-2Ra antibodies were used in the cultures. This autologous IL-2 production might explain the similarities in transcriptomes between TB IL-2 and TB nil (Figure 5B) and the small differences in gene expression. The failure to account for endogenous IL-2 production could greatly skew the authors' findings. Minimally, these controls should be provided at some point. Along the same line, the relatively modest effect of ruxolitinib compared to absence of exogenous IL-2 is surprising; more controls are needed to document the efficacy of this treatment. Also, why do the ruxolitinib-dependent accessible peaks exceed IL-2 peaks? (2900 vs 1000?). Additionally, activation of T cells and induction of IL-2RA is typically associated with downregulation of IL-7R - clearly there is an IL-7 signal; but the authors should include flow cytometry with controls, as readers may be confused.

The authors conclude that withdrawal of IL-2 results in loss of accessibility in 1000 genomic regions; however, it is unclear on what basis the 1000 peaks were selected. It is a little hard to imagine how the shared numbers of peaks turned out to be exactly 1000 peaks

It would be of interest for the authors to show expression of other Jun-family members? Would have been of interest to delete JunD by Crspr/Cas to identify contribution of JunD versus other family members (e.g. JunB), as the Ciofani laboratory has reported that JunB preferentially impacts Th17 differentiation (PMID:28824171)

Also of interest, would be to provided non-biased motif enrichments.

Since IL-2 induces Myc and Myc is an overall amplifier of gene expression in lymphocytes, the authors will need to take this into account in their analyses (Nie et al, PMID:23021216; PMID:23021215).

Minor points

Overall, the manuscript is hard to follow. The authors frequently switch back to discussion of their previous work in results sections. It would be clearer if the authors focused more on description of the experiments provided (including relevant controls) and compared the present results with previous results in the discussion. Figure 1 can be included in supplementary material. In terms of ease of interpreting the data, the controls of T blasts (T = 40 hrs and 70 hrs) should be shown in Figure 2.

Transcription factors that comprise AP-1 are certainly inducible; however, it is more accurate to think of STAT5 as a latent cytosolic TF that is activated - rather than inducible.

With respect to de novo enhancer formation, the authors should cite the work from the Glass and Natoli laboratories. With regard to the role of AP-1, the authors should cite the work from the Barski lab (Yukawa et al, PMID:31653690)

Referee #2:

General summary

Previous work by these authors has identified epigenetically primed sites (pDHS), in CD4 T cells. These are regions of the genome that are preserved in an open state, and which allow efficient access for inducible transcription factors when effector T cells and memory T cells re-encounter antigen. These pDHS are maintained by constitutively active transcription factors, however, also have binding sites for STAT5 and AP-1, transcription factors that are activated by IL-2.

In this study, Bevington et al. have investigated the role of the cytokine IL-2 in regulating and maintaining both pDHS (induced when naive CD4 T cells are differentiated into primed T-blast cells) and iDHS (which are triggered when primed CD4 T-blast cells are re-stimulated by PMA and ionomycin to mimic antigen-receptor signaling). Here they show that IL-2 is required to maintain a subset of pDHS induced by TCR activation, and link AP-1 and STAT5 binding at these sites to maintaining the open chromatin conformations. They also determined that IL-2-regulated pDHS control steady-state gene expression, and were important for keeping chromatin open at sites accessed by transcription factors that determine Th1, Th2 and Th17 lineage commitment. Moreover, they identified iDHS that were IL-2 dependent, and that many of these were near to IL-2-dependent pDHS, indicating potential co-operation between IL-2 pDHS and iDHS regulatory elements. Moreover, as these changes were regulated in a similar way by IL-7, it suggests that other cytokines of the common gamma chain family may substitute for IL-2.

This study is well controlled, well written and provides a fascinating insight into how the functions of IL-2 go beyond that of a simple mitogen and growth factor for T cells to control CD4 T cell fate decisions. This is important for understanding how the differentiation and plasticity of CD4 T cells is regulated, and provides new perspectives that IL-2 controls fate decisions not just by controlling the abundance of transcription factors and cytokine receptors, but also primes the cells to response to these signals. This is important to consider when designing and developing T cell based therapies.

There were no major concerns to be addressed, but the manuscript would be improved by clarifying the following minor points:

Minor points

- 1) For the DNAase I analysis, live cells were sorted before preparing samples. Please can the authors clarify/discuss any steps they took to ensure that in the IL-2 (or IL-7)-nil samples, the live cells that they were analyzing were not beginning to undergo any changes to chromatin associated with apoptosis resulting from the cytokine withdrawal. Additionally, could the data between samples be influenced by alterations in cell cycle progression as a result of IL-2/IL-7 removal?
- 2) In this study is that the authors used publicly available datasets as resources to interpret their findings. In Figure 2, they use a published ATAC-seq dataset for Th2 cells, and compared it to their DNAase I analysis. Can the authors please comment on how comparable these approaches are for drawing conclusions about open chromatin regions or any information that could be missed.
- 3) In Figure 3C (and Figure S2B), it would be less ambiguous to label the axis of graphs quantifying the western blots with 'Relative signal intensity' than 'Relative expression level', especially for the quantification of a phosphorylated species.
- 4) Relating to the authors conclusions that 'IL-2 iDHSs were not inducible in TN cells which had not

yet undergone re-programming or been exposed to IL-2', while this is evidenced in Figure 4C, their model would be strengthened by including the TN+ sample in the gene peaks depicted in Figure 4G.

5) In Figure 4G, for clarity, it would be helpful if the authors labelled which regions were pDHSs and iDHSs.

6) On page 8, the authors state 'Down-regulated genes included genes associated with T cell homeostasis which are normally associated with a cell signature more reminiscent of naive T cells, such as Tcf7, Ccr7 and Cd62l (Sell)'. Can the authors please clarify to what the down-regulation is referring?

7) In Figure 6G, it would help if the pDHSs and iDHSs were labelled in the figure to clarify which are linked to the changes in gene expression.

Non-essential suggestions

1) The authors mention that IL-2 can both create and erase pDHSs. For example, chromatin accessibility at the Sell locus increases in T-blasts in response to IL-2 deprivation, indicating that IL-2 maintains a closed conformation at this site. The differentiation of Th17 cells is inhibited by IL-2 and although the authors have shown that IL-2-dependent pDHSs overlap with STAT3 and ROR γ binding sites, I wondered if the authors had investigated whether IL-2 may inhibit the formation pDHS at other loci that are associated with Th17 differentiation, for example the IL-6Ra locus?

Miscellaneous

1) Possible typo at the end of the legend for Table S2 (Table written in bold after TB+ IL-2nil).

Referee #3:

In this manuscript by Bevington et al the authors build on their previous findings that established primed DNase Hypersensitive Sites (pDHS) in blasting T cells as well as in memory T cells. In the present study, the authors make additional discoveries:

A) they identify that IL-2 signaling is instrumental in the maintenance of primed accessible chromatin at regulatory elements in activated blasting T cells before these cells become committed to helper lineages and thus IL-2 signaling paves the way for lineage-defining factors.

B) They confirm that these pDHS are IL-2 dependent since they are drastically reduced in the absence of IL-2.

C) By using publicly available ATAC-seq datasets they further confirm the existence of these IL-2 dependent pDHS in vivo generated Th2 cells isolated from the lung. This is a nice addition since not only they show that these regulatory elements are relevant in in vivo settings but also the figure nicely confirms the comparable sensitivity of the DNase-seq and ATAC-seq obtained peaks.

D) In addition, they demonstrate that IL-7 signaling can generate similar pDHS, suggesting that not only IL-2 but also other cytokines at different settings/ developmental stages can regulate chromatin priming.

E) The authors show that the identified accessible regions are strongly bound by IL-2 inducible, pioneer factors such as JUND and STAT5 and in a detailed manner the authors dissect the IL-2 dependency of these binding events.

F) They also identify "inducible" DHS that are established upon activation of blasting cells with PMA and calcium ionophore.

G) They further dissect potential functional implications of the identified IL-2 dependent pDHS on gene expression. The role of IL-2 signaling is more impactful on steady state gene expression (in blasting cells) and less strong in inducible reactivation.

H) The authors also demonstrate that the IL2 pDHS of signature genes of T helper subsets can be bound by STAT factors and the respective determining factors of each subset.

The manuscript is well-written and the data seem to be overall of good quality. The figures are easy to follow. The topic is of broad interest since it provides novel insight on how environmental cues (cytokine signaling) can impact gene expression priming and act as an early event in opening chromatin.

Points to be addressed:

1) As mentioned above the addition of the in vivo Th2 ATACseq datasets is interesting and the confirmation of il-2 dependent pDHS in this setting is important. Have the authors attempted to identify similar pDHS in other in vivo generated cells such as T follicular cells (Tfh) or non-helper cells such as regulatory T cells for which ATAC-seq or both ATAC and DNase-seq datasets are available? It would be interesting to identify common versus lineage specific pDHS.

2) How many biological replicates are there for the ChIP-seq datasets? At least two biological replicates should be provided. It should also be clarified in the methods section.

3) In Fig. 5 it would be very helpful if the data of Fig. 5C were represented as an MA plot and the number of upregulated and downregulated genes were indicated.

4) In Fig 7A it is quite intriguing that the vast majority of the pDHS are shared between Th1 and Th17 cells. This is quite interesting since IL-2 plays opposing roles by promoting Th1 and inhibiting Th17 cell differentiation. Are these pDHS "active" in Th1 cells and non-active in Th17 cells? This could be assessed by histone marks (such as H3K27Ac) or binding of p300 or enrichment of 5hmC (potential available datasets can be found for example at Ciofani et al, Cell 2012 or Ichiyama et al, Immunity 2015).

Minor points:

There are some typos to be corrected or clarifications must be added in the Methods section.

1) Eg: page 18 Western blot section: the name of the product used for preparing nuclear extracts is missing

2) In the same section there is a question mark following the 2ndary antibody.

3) What is the dilution of the antibodies used in the Western blots?

4) Chromatin IP section: Chromatin was sheared (? Cells/ ml)

5) Library preparation: How many rounds of amplification were they performed?

Referee #1:

The authors conclusions are provocative and interesting; however, the study relies heavily upon in vitro stimulation using artificial conditions (concanavalin A, PMA/ionomycin etc). The authors do not test their hypotheses using solid in vivo systems, and in this respect, the scope of the manuscript is necessarily limited.

The idea that IL-2/STAT5 provide critical "pioneer" functions in chromatin accessibility and priming of gene expression is exciting; however, the impact of the findings would be greater if robust in vivo models were included.

As summarised above, we have now included data for both Th1 and Th17 cells generated *in vivo* in response to infection. Exploiting two different mouse models, these cells have been isolated to high purity. These data reveal stable epigenetic reprogramming of the same subset of primed DHSs that we showed are generated *in vitro* by TCR signalling and maintained by IL-2 or IL-7.

The mechanistic role of IL-2 in memory cells is an important topic; but the conclusions from the present study are largely inferential.

In our studies, we have extensively tested the role of IL-2 *in vitro*. We have also now included data that also specifically tests the role of IL-7 signals, which we think has overlapping roles with IL-2 in maintaining the epigenetic changes described. We think the inclusion of this new data addresses the reviewers concern and provides new direct mechanistic insight..

Activated T cells produce IL-2; this raises the issue as to whether the results might be more dramatic if anti-IL-2/anti-IL-2Ra antibodies were used in the cultures. This autologous IL-2 production might explain the similarities in transcriptomes between TB IL-2 and TB nil (Figure 5B) and the small differences in gene expression. The failure to account for endogenous IL-2 production could greatly skew the authors' findings. Minimally, these controls should be provided at some point.

To address this point in another way we have now included mRNA data in figure EV1A showing that IL2RA levels plummet when IL-2 is removed. We had already also addressed this issue by showing that IL-2-dependent remodelling substantially decreases when IL-2 signalling is inhibited by Ruxolitinib.

Any new analysis of IL-2 levels, or the inclusion of IL-2 antibodies is unlikely to change any of the interpretations of this study. At the level of DHSs, there already are major differences in the maintenance of IL-2-dependent DHSs when IL-2 is removed, so removing autocrine IL-2 is not likely to impact on this much further. For IL-2 to be effective, the cells need to express the inducible IL-2 R alpha gene.

Along the same line, the relatively modest effect of ruxolitinib compared to absence of exogenous IL-2 is surprising; more controls are needed to document the efficacy of this treatment. Also, why do the ruxolitinib-dependent accessible peaks exceed IL-2 peaks? (2900 vs 1000?).

Apart from the fact that the decrease is actually quite substantial, this comment does not take into account of the fact that there are more AP-1 sites than STAT sites in the IL-2-dependent DHSs. Ruxolitinib only inhibits JAK/STAT-dependent DHSs. Furthermore, FOS and JUN are strictly inducible whereas constitutive non-phosphorylated STAT5 can still bind at some DHSs (Figure 3E) and may still maintain a weak signal in the presence of Ruxolitinib. The discrepancy between 2900 versus 1000 is potentially confusing so we replaced the Venn diagram of larger subsets with an average profile over the 1000 IL-2-dependent DHSs, as figure EV1I, and this now more clearly shows the substantial decrease in average signal.

Additionally, activation of T cells and induction of IL-2RA is typically associated with downregulation of IL-7R - clearly there is an IL-7 signal; but the authors should include flow cytometry with controls, as readers may be confused.

In this study we aim to show that IL-2 and IL-7 can support similar functions in different contexts. We tend to think of IL-2 as the initial reprogramming signal, with IL-7 acting to maintain this signal once it is formed. For the purpose of this study, it is of less importance to distinguish which is acting at any one time. In the case of T blast cells we have now included data showing robust expression of IL-2RA (figure EV1A). We are not currently able to perform more FACS analyses.

The authors conclude that withdrawal of IL-2 results in loss of accessibility in 1000 genomic regions; however, it is unclear on what basis the 1000 peaks were selected. It is a little hard to imagine how the shared numbers of peaks turned out to be exactly 1000 peaks.

The 1000 IL-2 pDHSs were defined precisely as the overlap between the IL-2 dependent peaks defined in replicates 1 and 2. By chance this came to exactly 1000 DHSs which were both primed by T cell activation and IL-2 dependent.

It would be of interest for the authors to show expression of other Jun-family members? Would have been of interest to delete JunD by Crspr/Cas to identify contribution of JunD versus other family members (e.g. JunB), as the Ciofani laboratory has reported that JunB preferentially impacts Th17 differentiation (PMID:28824171)

We agree with the reviewer that this is an important point, and we have now created an additional figure 9 comparing the roles of JUNB and JUND, and highlighted their binding at *IL17a* and *Irf8* where these factors play key roles.

Figure 9A shows that JUNB expression in T blast cells in vitro is dependent on TCR signalling whereas JUND is dependent on IL-2 signalling. Figure 9B shows that binding of JUNB to IL-2-dependent DHSs is strictly dependent on TCR signalling.

We have now also made use of the data from the Ciofani paper (PMID:28824171) which showed that JUNB was needed for Th17 cells, and that it was upregulated during the Th17 differentiation process. Furthermore their data showed that in *Jun*-deficient cells, JUND binding increased at some of our IL-2-dependent pDHSs where JUNB was bound in the WT. Along these lines we have now included data showing that JUND binding in Th17 cells increases in the absence of JUNB (figures 9D,E). Using our own data we also show that expression of *Ccl1*, *IL-24* and *Csf2* mRNA expression actually increase in T cells from *Jund*-deficient mice (figure 9F). This is consistent with the published immune-suppressive properties on JUND. It may be a place holder maintaining homeostasis.

Also of interest, would be to provided non-biased motif enrichments.

It is unclear what the reviewer wants here. All of our motif analyses are unbiased de novo motif finding analyses.

Since IL-2 induces Myc and Myc is an overall amplifier of gene expression in lymphocytes, the authors will need to take this into account in their analyses (Nie et al, PMID:23021216; PMID:23021215).

It is possible that global mRNA levels change in the absence of IL-2. However, that is not the theme of this study. We have consistently shown relative differences between conditions, which is our goal.

Minor points

Overall, the manuscript is hard to follow. The authors frequently switch back to discussion of their previous work in results sections. It would be clearer if the authors focused more on description of the experiments provided (including relevant controls) and compared the present results with previous results in the discussion.

We feel it is important to guide the reader through the rationale of all the analyses by putting into a context as they are reading it, and not leave it all until the discussion. Independent readers of our manuscript requested this. Also consider that this is not a specialist journal, and context is important here.

Figure 1 can be included in supplementary material. In terms of ease of interpreting the data, the controls of T blasts (T = 40 hrs and 70 hrs) should be shown in Figure 2.

Figure one is actually essential to explain the background and methodology, especially for non-specialist readers. It sets the scene for the whole study. In our previous similar manuscripts we have been asked by reviewers to actually increase this amount of content to clarify how experiments were done.

Transcription factors that comprise AP-1 are certainly inducible; however, it is more accurate to think of STAT5 as a latent cytosolic TF that is activated - rather than inducible.

We are aware of this point, and we have shown data that shows STAT5 binding at some sites prior to activation.

With respect to de novo enhancer formation, the authors should cite the work from the Glass and Natoli laboratories.

We have now included a description in the discussion of parallel myeloid gene-specific patterns of regulation described by Natoli and Glass whereby genes undergo step-wise priming and can maintain an epigenetic memory of LPS stimulation.

With regard to the role of AP-1, the authors should cite the work from the Barski lab (Yukawa et al, PMID:31653690)

We have now cited Barski as the source for defining TCR/CD28-inducible transcription factors in activated T cells. This citation also includes data using dominant negative FOS which inhibits these pathways.

Referee #2:

General summary

There were no major concerns to be addressed, but the manuscript would be improved by clarifying the following minor points:

We are pleased to know that this reviewer found no major flaws with the manuscript.

Minor points

1) For the DNAase I analysis, live cells were sorted before preparing samples. Please can the authors clarify/discuss any steps they took to ensure that in the IL-2 (or IL-7)-nil samples, the live cells that they were analyzing were not beginning to undergo any changes to chromatin associated with apoptosis resulting from the cytokine withdrawal. Additionally, could the data between samples be influenced by alterations in cell cycle progression as a result of IL-2/IL-7 removal?

We have now included figure EV1B showing that Annexin V and PI staining do not change after IL-2 removal in the sorted live cells used here for our analyses.

Our new in vivo data is also based predominantly on non-dividing memory T cells which have a pattern similar to T blast cells dividing in vitro.

We have also highlighted that some of the genes that change are indeed cell cycle genes.

2) In this study is that the authors used publicly available datasets as resources to interpret their findings. In Figure 2, they use a published ATAC-seq dataset for Th2 cells, and compared it to their DNAase I analysis. Can the authors please comment on how comparable these approaches are for drawing conclusions about open chromatin regions or any information that could be missed.

By and large ATAC and DNaseI data are comparable. DNase I analyses provide a more accurate picture of open chromatin than ATAC which can also produce peaks in flanking regions between active nucleosomes. Figure 2A only shows overlap between the ATAC data where there is also a DNase-peak, so this is not an issue. Our new Th1 and Th17 data is also ATAC and has a 65% overlap with in vitro T blast DNaseI data.

3) In Figure 3C (and Figure S2B), it would be less ambiguous to label the axis of graphs quantifying the western blots with 'Relative signal intensity' than 'Relative expression level', especially for the quantification of a phosphorylated species.

We have corrected these figures.

4) Relating to the authors conclusions that 'IL-2 iDHSs were not inducible in TN cells which had not yet undergone re-programming or been exposed to IL-2', while this is evidenced in Figure 4C, their model would be strengthened by including the TN+ sample in the gene peaks depicted in Figure 4G.

We have included TN+ data in what is now figure 5G.

5) In Figure 4G, for clarity, it would be helpful if the authors labelled which regions were pDHSs and iDHSs.

We have now indicated in that the red boxes refer to iDHSs in what is now 5G.

6) On page 8, the authors state 'Down-regulated genes included genes associated with T cell homeostasis which are normally associated with a cell signature more reminiscent of naive T cells, such as Tcf7, Ccr7

and Cd62l (Sell).' Can the authors please clarify to what the down-regulation is referring?

The text was incorrect. The paragraph has been rewritten for clarity

7) In Figure 6G, it would help if the pDHSs and iDHSs were labelled in the figure to clarify which are linked to the changes in gene expression.

We have now highlighted the regions in what is now figure 7D.

Non-essential suggestions

1) The authors mention that IL-2 can both create and erase pDHSs. For example, chromatin accessibility at the Sell locus increases in T-blasts in response to IL-2 deprivation, indicating that IL-2 maintains a closed conformation at this site. The differentiation of Th17 cells is inhibited by IL-2 and although the authors have shown that IL-2-dependent pDHSs overlap with STAT3 and ROR γ binding sites, I wondered if the authors had investigated whether IL-2 may inhibit the formation pDHS at other loci that are associated with Th17 differentiation, for example the IL-6Ra locus?

This is getting beyond the scope of this manuscript.

Miscellaneous

1) Possible typo at the end of the legend for Table S2 (Table written in bold after TB+ IL-2nil).

We have corrected this

Referee #3:

The manuscript is well-written and the data seem to be overall of good quality. The figures are easy to follow. The topic is of broad interest since it provides novel insight on how environmental cues (cytokine signaling) can impact gene expression priming and act as an early event in opening chromatin.

Points to be addressed:

1) As mentioned above the addition of the in vivo Th2 ATACseq datasets is interesting and the confirmation of il-2 dependent pDHS in this setting is important. Have the authors attempted to identify similar pDHS in other in vivo generated cells such as T follicular cells (Tfh) or non-helper cells such as regulatory T cells for which ATAC-seq or both ATAC and DNase-seq datasets are available? It would be interesting to identify common versus lineage specific pDHS.

We have now included data for Th1, Th2, Th17, Treg and Tfh cells in figures 2 C and D.

2) How many biological replicates are there for the ChIP-seq datasets? At least two biological replicates should be provided. It should also be clarified in the methods section.

We have now included the replicates for STAT5 and JUND ChIP-Seq from T blasts cells growing in IL-2. These are the pivotal data sets used to show that lineage-determining factors reoccupy these sites in Th1, 2 and 17 cells.

With the agreement of the editor, we have include just one ChIP-seq data set for IL-2 nil cells. It was deemed non-essential to have a replicate on the basis that STAT5 and JUND were unlikely to bind if the actual DHS was lost, as confirmed in two DNase-Seq replicates.

3) In Fig. 5 it would be very helpful if the data of Fig. 5C were represented as an MA plot and the number of upregulated and downregulated genes were indicated.

We have now included an MA plot as figure 6C, and we believe that we addressed the numbers on p8 with the statements:

“the presence of IL-2 had the biggest impact on gene expression prior to stimulation (T_B IL-2 compared to T_B IL-2nil Figure 6B) with 612 genes being more than 3 fold up-regulated in T_B IL-2 compared to T_B IL-2nil and 356 genes that were at least 3 fold down-regulated when IL-2 was present (adj-p <0.05) (Figure 6C, Supplemental data file 2 S1-2).”

And

“Removal of IL-2 in T_B+ IL-2nil cells affected the induction of just 14 genes by at least two-fold (Supplemental data file 2).”

4) In Fig 7A it is quite intriguing that the vast majority of the pDHS are shared between Th1 and Th17 cells. This is quite interesting since IL-2 plays opposing roles by promoting Th1 and inhibiting Th17 cell differentiation. Are these pDHS "active" in Th1 cells and non-active in Th17 cells? This could be assessed by histone marks (such as H3K27Ac) or binding of p300 or enrichment of 5hmC (potential available datasets can be found for example at Ciofani et al, Cell 2012 or Ichiyama et al, Immunity 2015).

We have added a new figure (Figure 9) with published ChIP-seq data (Carr et al, Vahedi et al) to show examples of sites which are shared between Th1 and Th17 cells and how they may be differentially regulated. At the signature Th17 gene, *Il17a*, an IL-2 sensitive DHS is shared between Th1 and Th17 cells, but p300 only binds to the site in Th17 cells. Conversely at the *Irf8* locus which is a known repressor of Th17 cells, the IL-2 pDHSs binds p300 only in Th1 cells. Furthermore, the data suggest that JUNB regulates *Il17a* expression in Th17 cells and JUND represses it, whilst the reciprocal is inferred for *Irf8*.

Minor points:

There are some typos to be corrected or clarifications must be added in the Methods section.

1) Eg: page 18 Western blot section: the name of the product used for preparing nuclear extracts is missing

We have corrected this

2) In the same section there is a question mark following the 2ndary antibody.

We have added this information

3) What is the dilution of the antibodies used in the Western blots?

We have added this information

4) Chromatin IP section: Chromatin was sheared (? Cells/ ml)

We have corrected this

5) Library preparation: How many rounds of amplification were they performed?

We have added this information

Dear Peter,

Thanks for submitting your revised manuscript to The EMBO Journal. Your manuscript has now been seen by referees # 2 and 3 and their comments are provided below. As you can see both referees support publication here. Referee #2 has a few relative minor comments that I would like to ask you to address in a final revision.

When you submit the revised version will you also please take care of the following points:

- We need 3-5 keywords
- Please re-label "Declaration of Interests" as Conflict of interest
- The Data availability section should be listed after conflict of interest. Will you also make sure that you make the data available and remove the password protection
- David Withers needs an ORCID
- The figures should be uploaded as separate files and another format than powerpoint.
- = Please add the legends for the EV tables 1 and 2 to the tables as a separate tab
- For the reference format please add et al after the 10th author
- The synopsis image should be uploaded as a TIFF file.
- Some of the same images are displayed in 2A, 3D, 6D, EV1C, EV2A can you please mention this in the figure legends.
- Our publisher has also done their pre-publication check on your manuscript. When you log into the manuscript submission system you will see the file "Data edited manuscript file". Please take a look at the word file and the comments regarding the figure legends and respond to the issues. Please also use this version when you resubmit the revised version with the marked changes. Just makes it easier for me to see the changes.

That should be all - let me know if you have any further questions

With best wishes

Karin

Karin Dumstrei, PhD
Senior Editor
The EMBO Journal

- a point-by-point response to the referees' comments, with a detailed description of the changes made (as a word file).

- a word file of the manuscript text.

- individual production quality figure files (one file per figure)

- a complete author checklist, which you can download from our author guidelines

(<https://www.embopress.org/page/journal/14602075/authorguide>).

- Expanded View files (replacing Supplementary Information)

Further information is available in our Guide For Authors:

The revision must be submitted online within 90 days; please click on the link below to submit the revision online before 3rd Nov 2020.

Link Not Available

Referee #2:

I thank the authors for their response to my comments, which have been addressed satisfactorily.

The addition of more in vivo data to the study provides evidence that regions of open chromatin in in vivo populations of CD4 T cells overlap with the IL-2-regulated pDHSs identified in the in vitro experiments. This indicates that there is a potential for IL-2 to regulate these sites in vivo. However, the data is correlative, and does not necessarily add substantial further evidence that IL-2 signaling pioneers CD4 T cell differentiation.

The addition of the in vivo data where the IL-7 α ; chain was knocked-out of memory CD4 T cells

does support a role for IL-7 signaling in maintaining regions of open chromatin in memory T cells in vivo.

I have a few minor comments:-

1) Can the authors please comment on survival of the CD4 memory T cells in their in vivo model once the IL-7R α has been deleted. Were similar numbers of IL-7 α ⁺ and IL-7 α ⁻ CD4 T cells isolated from Il7rf/f mice and CD4creERT2 x Il7rf/f mice respectively?

2) It may be helpful, particularly for ease of understanding of this work by a non-specialized audience, to discuss the IL-2 data, and then have the new IL-7 data as a section at the end of the manuscript rather than its current location. I think that would help to clarify the concept of IL-2 supporting the initial reprogramming, with IL-7 being involved in longer-term maintenance. Otherwise, this point may become a little lost.

3) Along the same lines, it may be beneficial to modify the title of the manuscript to reflect the authors aim to demonstrate that IL-2 and IL-7 both support chromatin remodelling, but in different contexts. The addition of the in vivo data with the deletion of IL-7R α in CD4 memory T cell takes the focus of the paper beyond the role of IL-2 alone.

4) The new section discussing the findings that IL-2-induced pDHS are shared between Th lineages is very interesting, particularly that maintaining these open chromatin regions may contribute to CD4 T cell plasticity. The authors suggest that Th17 cell differentiation may be inhibited by IL-2 through JUND binding to these sites to inhibit transcription. While this may be a contributing factor, I do think that it is important that the authors include a comment in the text that acknowledges that other factors, such as IL-2 decreasing chromatin accessibility at certain genomic loci, could integrate with these factors to contribute to IL-2-induced inhibition of Th17 differentiation.

Miscellaneous

1) Typo 'diffentiation' on page 12.

Referee #3:

The authors have addressed in depth my comments. In addition, in the revised manuscript they have included novel, very interesting in vivo data that strengthen significantly the original conclusions. Overall, the revised manuscript is suitable for publication.

Response to reviews and editors comments

Requests from the editor:

- We need 3-5 keywords
Added to title page

- Please re-label "Declaration of Interests" as Conflict of interest
Done

- The Data availability section should be listed after conflict of interest. Will you also make sure that you make the data available and remove the password protection
The section is moved.
The GEO is being prepared now for public release as soon as possible.

- David Withers needs an ORCID
The ORCID number is 0000-0003-3757-7594
I cannot add it myself.
I have asked David to create an account.

- The figures should be uploaded as separate files and another format than powerpoint.
Files added as PDF

-Please add the legends for the EV tables 1 and 2 to the tables as a separate tab
Done

- For the reference format please add et al after the 10th author
New format changed to single 1st au in text and et al after 10 au in list

- The synopsis image should be uploaded as a TIFF file.
Uploaded as TIFF

- Some of the same images are displayed in 2A, 3D, 6D, EV1C, EV2A can you please mention this in the figure legends.
We have referred back to figure 2A for each legend

Referee 2 comments:

1) Can the authors please comment on survival of the CD4 memory T cells in their in vivo model once the IL-7R α has been deleted. Were similar numbers of IL-7 α ⁺ and IL-7 α ⁻ CD4 T cells isolated from Il7rf/f mice and CD4creERT2 x Il7rf/f mice respectively?

We created a new panel showing a 2-fold decrease in memory T cells after deleting IL-7R α

2) It may be helpful, particularly for ease of understanding of this work by a non-specialized audience, to discuss the IL-2 data, and then have the new IL-7 data as a section at the end of the manuscript rather than its current location. I think that would help to clarify the concept of IL-2 supporting the initial reprogramming, with IL-7 being involved in longer-term maintenance. Otherwise, this point may become a little lost.

The IL-7 data has been extracted from figure 2 and EV1 and moved to a new figure 7.
The IL-7R α data is now in Figure 8 and EV4.

3) Along the same lines, it may be beneficial to modify the title of the manuscript to reflect

the authors aim to demonstrate that IL-2 and IL-7 both support chromatin remodelling, but in different contexts. The addition of the in vivo data with the deletion of IL-7R α in CD4 memory T cell takes the focus of the paper beyond the role of IL-2 alone.
The title was increased to 109 characters plus spaces to include IL-7

4) The new section discussing the findings that IL-2-induced pDHS are shared between Th lineages is very interesting, particularly that maintaining these open chromatin regions may contribute to CD4 T cell plasticity. The authors suggest that Th17 cell differentiation may be inhibited by IL-2 through JUND binding to these sites to inhibit transcription. While this may be a contributing factor, I do think that it is important that the authors include a comment in the text that acknowledges that other factors, such as IL-2 decreasing chromatin accessibility at certain genomic loci, could integrate with these factors to contribute to IL-2-induced inhibition of Th17 differentiation.

We have commented on the roles of STAT3, STAT5 and IL-6R α in Th17 differentiation

Miscellaneous

1) Typo 'diffentiation' on page 12.

Corrected

Dear Peter,

Thanks for submitting your revised manuscript to The EMBO Journal. I have now had a chance to take a careful look at the manuscript at the introduced changes and all looks good.

I am therefore very happy to accept the manuscript for publication here. Please make sure to make the GEO deposited data available upon publication of the study.

Congratulations on a very nice study!

With best wishes

Karin

Please note that it is EMBO Journal policy for the transcript of the editorial process (containing referee reports and your response letter) to be published as an online supplement to each paper. If you do NOT want this, you will need to inform the Editorial Office via email immediately. More information is available here: http://emboj.embopress.org/about#Transparent_Process

Your manuscript will be processed for publication in the journal by EMBO Press. Manuscripts in the PDF and electronic editions of The EMBO Journal will be copy edited, and you will be provided with page proofs prior to publication. Please note that supplementary information is not included in the proofs.

Should you be planning a Press Release on your article, please get in contact with embojournal@wiley.com as early as possible, in order to coordinate publication and release dates.

If you have any questions, please do not hesitate to call or email the Editorial Office. Thank you for your contribution to The EMBO Journal.

Corresponding Author Name: Peter Cockerill

Journal Submitted to: EMBO J

Manuscript Number: EMBOJ-2020-105220